# Predictive biomarkers for 5-fluorouracil and oxaliplatin-based chemotherapy in gastric cancers via profiling of patient-derived xenografts

Deukchae Na[1,15], Jeesoo Chae [2,3,15], Sung-Yup Cho[2,4,5,15], Wonyoung Kang[6], Ahra Lee[7], Seoyeon Min[7], Jinjoo Kang[7], Min Jung Kim[5], Jaeyong Choi[2], Woochan Lee [2], Dongjin Shin[2], Ahrum Min[4,8], Yu-Jin Kim[4], Kyung-Hun Lee[4,9], Tae-Yong Kim[4,9], Yun-Suhk Suh[10,11], Seong-Ho Kong[10], Hyuk-Joon Lee[4,10], Woo-Ho Kim [4,12], Hansoo Park [13], Seock-Ah Im[4,9✉], Han-Kwang Yang[4,10✉], Charles Lee[6,7,14✉] & Jong-Il Kim [2,4,5✉]

Gastric cancer (GC) is commonly treated by chemotherapy using 5-fluorouracil (5-FU) derivatives and platinum combination, but predictive biomarker remains lacking. We develop patient-derived xenografts (PDXs) from 31 GC patients and treat with a combination of 5-FU and oxaliplatin, to determine biomarkers associated with responsiveness. When the PDXs are defined as either responders or non-responders according to tumor volume change after treatment, the responsiveness of PDXs is significantly consistent with the respective clinical outcomes of the patients. An integrative genomic and transcriptomic analysis of PDXs reveals that pathways associated with cell-to-cell and cell-to-extracellular matrix interactions enriched among the non-responders in both cancer cells and the tumor microenvironment (TME). We develop a 30-gene prediction model to determine the responsiveness to 5-FU and oxaliplatin-based chemotherapy and confirm the significant poor survival outcomes among cases classified as non-responder-like in three independent GC cohorts. Our study may inform clinical decision-making when designing treatment strategies.

[1] Ewha Institute of Convergence Medicine, Ewha Womans University Mokdong Hospital, Seoul, Korea. [2] Department of Biomedical Sciences, Seoul National University College of Medicine, Seoul, Korea. [3] Cancer Evolution Research Center, The Catholic University of Korea, Seoul, Korea. [4] Cancer Research Institute, Seoul National University, Seoul, Korea. [5] Medical Research Center, Genomic Medicine Institute (GMI), Seoul National University, Seoul, Korea. [6] The Jackson Laboratory for Genomic Medicine, Farmington, CT, USA. [7] Department of Life Science, Ewha Womans University, Seoul, Korea. [8] Biomedical Research Institute, Seoul National University Hospital, Seoul, Republic of Korea. [9] Department of Internal Medicine, Seoul National University Hospital, Seoul National University College of Medicine, Seoul, Korea. [10] Department of Surgery, Seoul National University College of Medicine, Seoul, Korea. [11] Department of Surgery, Seoul National University Bundang Hospital, Seoul, Korea. [12] Department of Pathology, Seoul National University College of Medicine, Seoul, Korea. [13] Department of Biomedical Science and Engineering, Gwangju Institute of Science and Technology (GIST), Gwangju, Korea. [14] Precision Medicine Center, The First Affiliated Hospital of Xiu'an Jiaotong University, Shaanxi, People's Republic of China. [15] These authors contributed equally: Deukchae Na, Jeesoo Chae, Sung-Yup Cho. ✉email: moisa@snu.ac.kr; hkyang@snu.ac.kr; Charles.Lee@jax.org; jongil@snu.ac.kr

Gastric cancer (GC) is responsible for >1,000,000 new cases and an estimated 783,000 deaths worldwide in 2018, making GC the fifth most frequently diagnosed cancer and the third leading cause of cancer death[1]. Although progress has been made in understanding GC etiology, GC treatments continue to fail in many patients. Curative surgery followed by adjuvant chemotherapy or chemoradiotherapy are standard-of-care treatments for locoregional GCs[2]. More than 50% of patients undergo surgery; however, even after curative resection, approximately 60% of patients relapse locally or with distant metastases[3]. Folinic acid, 5-fluorouracil (5-FU), and oxaliplatin (FOLFOX) and capecitabine (5-FU prodrug) and oxaliplatin (XELOX) are widely used chemotherapy regimens for GC treatment[4]. However, responsiveness of these regimens was various among patients and several cases have reported multidrug resistance after treatment[5].

Genomic and transcriptomic profiling can provide rich sources of data for patient classification. The molecular subtyping of GC has progressed substantially during the past decade, especially by The Cancer Genome Atlas (TCGA)[6] and the Asian Cancer Research Group (ACRG)[2]. ACRG classified GC into four molecular subtypes: microsatellite-instable (MSI), microsatellite-stable with epithelial–mesenchymal transition (EMT) expression (MSS/EMT), microsatellite-stable with activated TP53 (MSS/TP53$^+$), and microsatellite-stable with TP53 functional loss (MSS/TP53$^-$). Patient prognosis was best for MSI and worst for MSS/EMT[2]. However, these classifications are not applicable for predicting responsiveness to standard chemotherapy. The Singapore-Duke study identified three molecular GC subtypes, based on expression patterns: proliferative, metabolic, and mesenchymal[7]. Metabolic subtype tumors appeared to display enhanced benefits in response to 5-FU treatment. However, these subtype classifications remain insufficient and are inappropriate predictors for chemotherapy responsiveness.

Patient-derived xenografts (PDXs) are widely used model systems that recapitulate the histopathological characteristics, molecular characteristics, and drug responses of parental tumors. We extensively investigated potential predictive biomarkers for the responsiveness to 5-FU and oxaliplatin-based chemotherapy in GC, which is one of the most common chemotherapy regimens, by combining genomic/transcriptomic analyses and in vivo drug responsiveness data with PDX models. PDX models showed a similar responsiveness to the clinical responses in matched GC patients, and convergent pathway alterations of both tumor and microenvironment were associated with treatment resistance. This study provided a valuable resource, with clinical utility, for evaluating the responsiveness to 5-FU and oxaliplatin-based chemotherapy and therapeutic decision-making in GC patients.

## Results
### Estimation of responsiveness to 5-FU and oxaliplatin-based chemotherapy using GC PDX models.
To investigate potential predictive factors associated with chemotherapy responsiveness in GCs, we established a GC PDX cohort, consisting of 32 PDX cases from 31 GC patients, and performed in vivo screening for 5-FU and oxaliplatin-based chemotherapy (Fig. 1a and detailed clinical information in Supplementary Table 1, age: 40–86 years, sex: 25 males and 6 females). PDX tumors retained driver mutations and genomic/characteristics of the original patient tumors [median percentage of shared somatic mutations was 67% between patient and PDX tumors, median Pearson's correlation coefficient of somatic mutation allele frequency was 0.72 ($P = 0.00016$), and median Pearson's correlation coefficient of mRNA expression was 0.56 ($P = 0.005$)]. When the mutations with therapeutic importance were considered, mutations in genes

applicable to Food and Drug Administration-approved drugs and standard care (OncoKB level 1 and 2; http://www.oncokb.org) were maintained in all of the patient and PDX pairs (Supplementary Fig. 1 and Supplementary Data 1). Clonal structures, estimated using the PyClone algorithm[8], were well conserved between patient and PDX tumors (Supplementary Fig. 2a, b). Because FOLFOX is a widely used standard chemotherapy regimen among GC patients, we treated the GC PDX models with 5-FU + oxaliplatin. The PDX models were classified into 13 responders (Rs), 11 non-responders (NRs), and 8 cases with questionable responsiveness (Fig. 1a, b). Rs were classified by comprehensive evaluation in that it reflects the tumor volumes before and after treatment with both drug- and vehicle-treated samples using the following criteria: (1) drug treatment significantly inhibited tumor growth [two-way analysis of variance (ANOVA) $P < 0.0001$; Supplementary Fig. 3a–c and Supplementary Table 2]; and (2) the percent tumor growth inhibition (TGI (%); described in "Methods") by drug treatment at end point was >60% (Fig. 1b and Supplementary Table 2)[9]. The average TGI was 81.0 and 22.9% in Rs and NRs, respectively, with a significant difference ($U$ test $P < 0.0001$). However, tumor growth rates between the R and NR groups did not differ significantly during the initial xenotransplantation stage (Passage 0; Supplementary Fig. 4a) or in vehicle-treated mice (Passage 2; Supplementary Fig. 4b), suggesting that tumor growth rate had little effect on the responsiveness classification.

Next, we compared responsiveness to 5-FU and oxaliplatin in PDX models with clinical responsiveness in matched patients. Among 10 patients who were treated with a comparable regimen, 3 of the 4 patients whose PDX models were classified into the R group showed clinically relevant responsiveness to 5-FU and platinum-based regimens (median progression-free survival time >46 months; Fig. 1c, d). Representatively, patient #3 received 54 cycles of XELOX (capecitabine plus oxaliplatin) treatment and showed no signs of disease progression for 51 weeks (Fig. 1d). In contrast, all 6 patients whose PDX models were classified as NR exhibited treatment resistance to 5-FU + platinum-based regimens, resulting in relapse or tumor progression (median progression-free or disease-free survival time of 5 months, ranging from 1 to 25 months; Fig. 1c, d). Drug responsiveness in the PDX models was significantly concordant with recurrence in matched patients ($P = 0.033$, Chi-square test), and patients classified as NR, based on the PDX response, showed poorer prognosis than patients classified as R, based on progression-free survival ($P = 0.0212$, log-rank test; Fig. 1e). These data suggested that responsiveness to 5-FU and oxaliplatin-based chemotherapy could be reliably predicted according to the PDX model criteria, anticipating the clinical responsiveness in patients.

### Investigation of previously reported factors associated with 5-FU responsiveness in our PDX cohort.
First, we compared clinical characteristics between R and NR patients, classified based on the PDX model response. Several clinical parameters, including gender, age, World Health Organization and Lauren classifications, tumor location, and tumor stage [Tumor, Node, Metastasis (TNM)], were not associated with responsiveness to 5-FU and oxaliplatin (Supplementary Table 3). Additionally, based on molecular profiling, TCGA molecular GC subtypes[6] were not associated with drug responsiveness ($P = 0.464$; Supplementary Table 4).

Previous studies suggested that genes in the one-carbon metabolism pathway[10], microsatellite instability[11], and p53 status[12] were associated with the 5-FU clinical response, for many cancer types. The one-carbon metabolism pathway includes methylenetetrahydrofolate reductase (MTHFR),

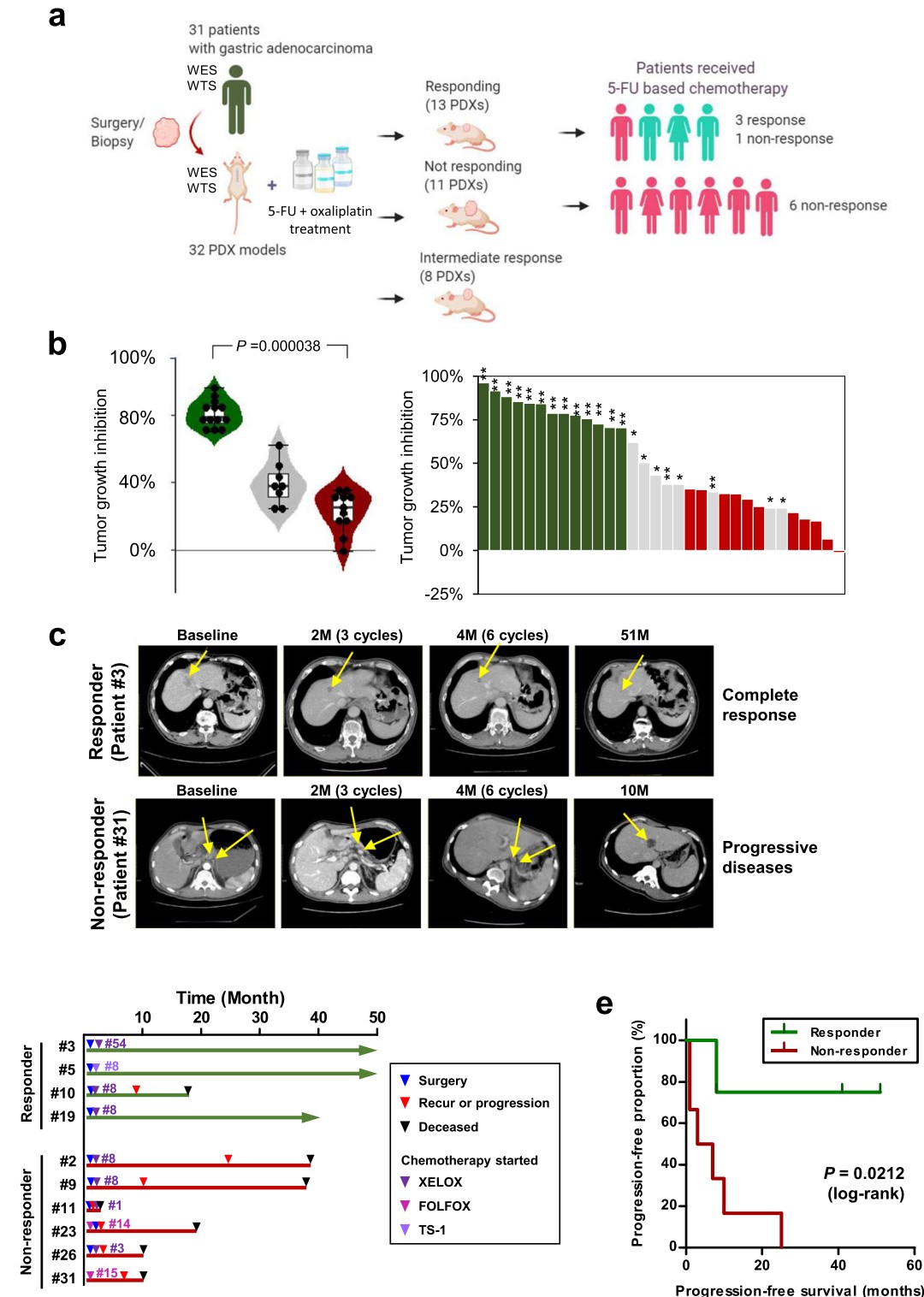

methionine synthase (*MTR*), methionine synthase reductase (*MTRR*), and thymidylate synthase (*TYMS*)[10]. Single-nucleotide polymorphisms (SNPs) and gene expression levels were suggested to serve as predictive markers for 5-FU-based chemotherapy responsiveness[10]. However, no differences were observed in germline and somatic mutation status (Supplementary Fig. 5a, b) or one-carbon metabolism-associated gene expression levels (Supplementary Fig. 5c) between the R and NR groups in our study.

Microsatellite instability results from the inactivation of mismatch repair (MMR) pathways and the benefit from FOLFOX chemotherapy with MSI tumors remains controversial[11,13]. When we examined MMR pathway genes, recurrent somatic mutations in these genes were not associated with 5-FU + oxaliplatin treatment responsiveness among our PDX cohort (Supplementary Fig. 6a). The expression levels of *MLH1*, which is one of the essential MMR genes, was not significantly different between the R and NR groups (Supplementary Fig. 6b). However, mutation

**Fig. 1 Classification of gastric cancer (GC) PDX models, based on responsiveness to 5-FU-based chemotherapy, and similarities to patients' clinical responses. a** Graphical overview of the study. This figure was generated using BioRender. WES whole-exome sequencing, WTS whole-transcriptome sequencing. **b** Percent tumor growth inhibition (TGI (%)) of 32 PDX models. Asterisks indicate significant differences between the responder and non-responder groups (*$P < 0.05$; **$P < 0.0001$). Violin plot shows the TGI values of PDX models of the responder ($n = 13$), non-responder ($n = 11$), and intermediate-responder ($n = 8$) groups. In the box and whisker plots, data are presented as median value and standard deviation, and the bottom and top edges of the box indicate the 25th and 75th percentiles, respectively. Two-tailed Mann–Whitney $U$ test using normal distribution ($P = 0.000038$) was performed. **c** Computed tomographic images of two representative patients with clinical responses that were consistent with their respective PDX responses. Cycles of chemotherapy are indicated in the bracket. The yellow arrow indicates tumor mass. M months. **d** Clinical outcome of patients treated with 5-FU-based chemotherapy. Of four patients whose PDX models were classified as responders, three patients (patient #3, #5, and #19) showed prolonged survival, without recurrence or progression to XELOX and TS-1 treatment. All of the six patients whose PDX models were classified as non-responders showed tumor progression and poor prognosis. XELOX capecitabine (prodrug of 5-FU) and oxaliplatin, FOLFOX folinic acid, 5-FU, and oxaliplatin, TS-1 a combination of tegafur, gimeracil, and oteracil potassium. TS-1 is converted into 5-FU after absorption. The number of treatment cycles is indicated after the drug names. **e** Progression-free survival of patients, based on the responses of their respective PDX models. Two-sided log-rank test ($P = 0.0212$).

burden was highly correlated with defective MMR signature, and *MLH1* expression showed significant anti-correlation with mutation burden and defective MMR signature (Supplementary Fig. 6c).

Functional p53 protein status has been suggested to affect 5-FU-based chemotherapy sensitivity in several cancer types[12,14,15]. Although not significantly, the somatic mutations frequencies in *TP53* increased in the NR group relative to the R group (38% for the R group and 64% for the NR group; Supplementary Fig. 7a, b), and the average p53 activity score was higher in the R group (0.19) than in the NR group (−0.11, Supplementary Fig. 7c). In addition, NRs with no *TP53* mutation also showed low p53 activity (group average: −0.13; Supplementary Fig. 7c), suggesting defective p53 function among the NR group. Although various factors associated with the MMR process and the p53 pathway were associated with the R or NR groups, previously suggested candidate predictive factors for 5-FU-based treatment did not sufficiently explain drug responsiveness in our GC PDX cohort, suggesting the need for additional treatment responsiveness biomarkers to be developed.

**Comprehensive genomic characterization of Rs and NRs to 5-FU and oxaliplatin-based chemotherapy.** To identify novel genomic features that separate the R and NR groups, we analyzed the genomic profiles of PDX tumors, using whole-exome sequencing (WES). Difference in the frequency of MSI-high samples (6/13 in Rs, 3/11 in NRs, $P = 0.597$) and tumor mutation burden (TMB; $P = 0.254$, when the blood DNA unmatched samples were excluded) were not significant between groups (Fig. 2a). In GC-associated cancer genes, suggested by the TCGA reports[6], no genes with significantly different mutation frequencies were identified between the R and NR groups (Fig. 2a). When comparing whole-exome mutation profiles between the R and NR groups, we identified 11 exclusively mutated cancer-associated genes, including *KIAA1549* (4/13), *BRD3* (3/13), *ZMYM2* (3/13), *PDGFRA* (3/13), *ICE1* (3/13), and *TRIM33* (3/13), in the R group, and *BCL9* (4/11), *HEY1* (3/11), *FOXO3* (3/11), *NCOR2* (3/11), and *DDR2* (2/11), in the NR group (Fisher's exact test, $P < 0.1$, Fig. 2a). The functional changes and clinical implications of these mutations remain to be further clarified.

Mutational signature analyses, based on mutations associated with single base substitutions (SBSs; https://cancer.sanger.ac.uk/cosmic/signatures/SBS/index.tt)[16], revealed that mutational signatures associated with defective MMR (SBS6, SBS14, SBS15, SBS20, SBS21, and SBS26) were higher in the R group (average proportion: 29.2% for Rs, 21.6% for NRs; Fig. 2b), whereas mutational signatures associated with age at cancer diagnosis (clock-like; SBS1, SBS5) were higher in the NR group (average proportion: 31.1% for Rs, 35.4% for NRs; Fig. 2b). A possible 5-

FU-associated signature (SBS17b) was observed in both the groups (average proportion: 2.6% for Rs, 2.2% for NRs; Fig. 2b). Analyses of small insertion and deletion (ID) signatures also showed a higher proportion of defective MMR signatures (ID1, ID2, and ID7) in the R group than in the NR group (Fig. 2c). The average proportions of ID2 were 33.2% for Rs and 27.1% for NRs and the average proportions of ID7 were 6.0% for Rs and 2.5% for NRs (Fig. 2c). Although the etiology remains unknown, ID9 was higher in the NR group than in the R group (average proportion: 8.8% for Rs and 17.4% for NRs; Fig. 2c). We also estimated the number of clonal mutation clusters, using PyClone[8]. The complexity of the clonal architecture was not significantly different between the groups (the median prevalence of changes between the patient and PDXs tumor were 0.22 and 0.16 in the R and NR groups, respectively; Supplementary Fig. 2c, d).

Next, we estimated somatic copy number alterations (SCNAs), using WES data, and compared copy number changes between groups. The amplification of 2q and the deletions of 16p and 19p were significantly enriched in the NR group, whereas the deletion of 11p was enriched in the R group ($P < 0.1$; Supplementary Fig. 8a). The percentages of the PDX tumor genome affected by SCNAs were not significantly different between the two groups (median percentage of genome altered for Rs: 17.1%, NRs: 14.2%, Supplementary Fig. 8b). Among the GC-associated cancer genes identified in the TCGA reports[6], the focal gene amplification of *CCND1* (23% for Rs, 0% for NRs), *MET* (38% for Rs, 18% for NRs), and *MDM2* (23% for Rs, 0% for NRs) was observed in the R group (Fig. 2d). *H3F3A*, which encodes histone protein H3.3A, was highly amplified in the R group (62% for Rs; 18% for NRs; Fig. 2d). In the NR group, the recurrent amplifications of *ETV1*, *PLAG1*, and *MAF* were significantly enriched compared with the R group (Fig. 2d). The functional changes and clinical implications of these alterations must be further clarified.

**Metabolic pathways define Rs, and cell-to-cell interactions define NRs to 5-FU and oxaliplatin-based chemotherapy.** We investigated the transcriptomic signatures of the R and NR groups by analyzing RNA sequencing data from PDX tumors. R and NR tumors were not easily discernible according to overall expression pattern due to mixed heterogeneous expression patterns, without distinct clustering patterns (Supplementary Fig. 9a). This indicates that drug responsiveness is not largely reflected by general gene expression patterns; instead, specific markers need to be examined through comparisons of the R and NR groups.

Gene set enrichment analysis (GSEA; Fig. 3a) and pathway analyses, based on differentially expressed genes (DEGs; Supplementary Fig. 9b, c) in the PDX transcriptome, revealed the enrichment of common gene sets associated with cancer

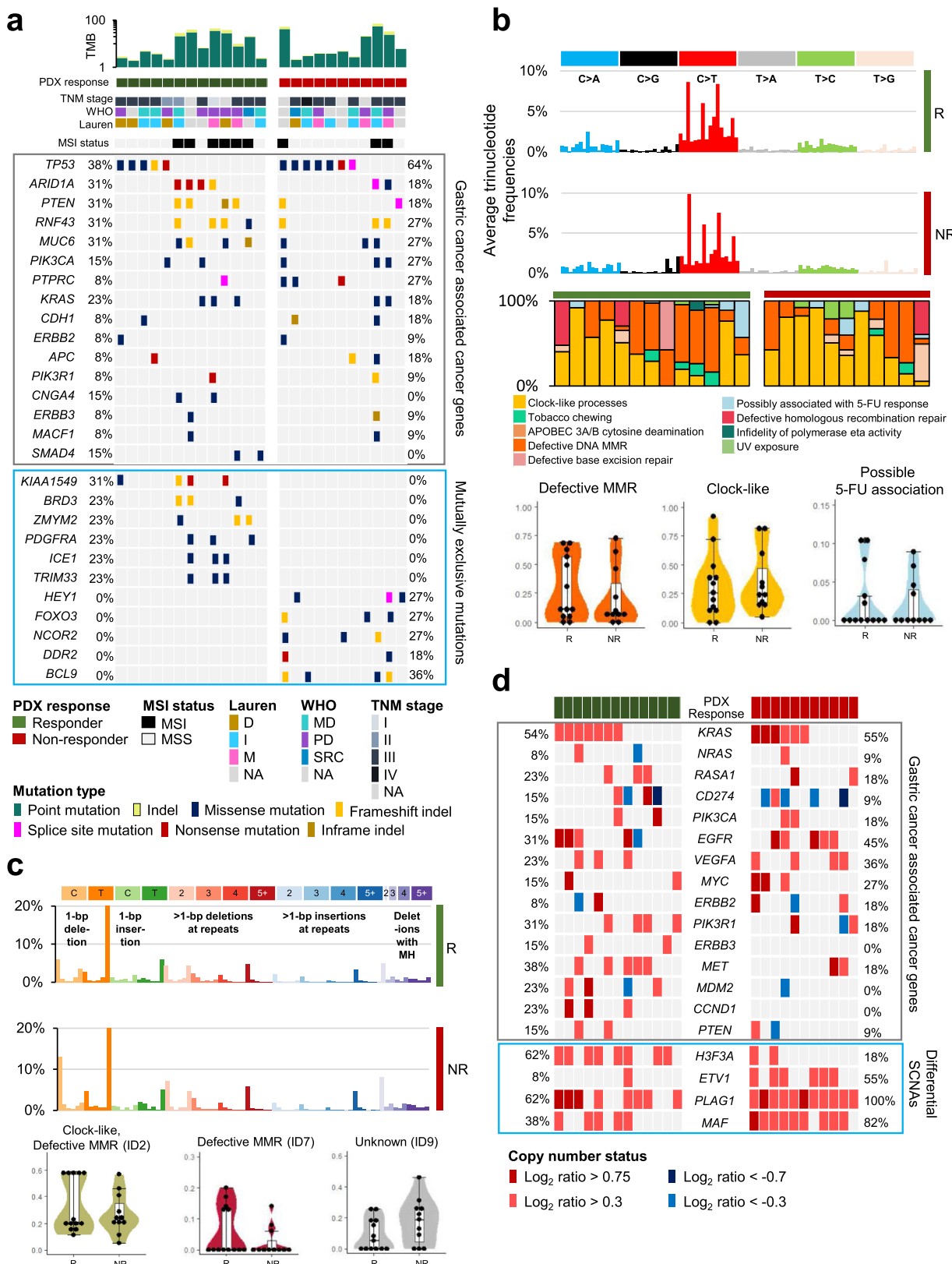

metabolism, including "REACTOME: mitochondrial fatty acid beta-oxidation," "KEGG: metabolism of xenobiotics by cytochrome P450," and "KEGG: biosynthesis of unsaturated fatty acids," in the R group (Fig. 3a and Supplementary Fig. 9b). In contrast, the NR group was enriched in gene sets associated with cell-to-cell and cell-to-extracellular matrix (ECM) interactions,

including "REACTOME: MET activates PTK2 signaling," "HALLMARK: Wnt/β-catenin signaling," "KEGG: ECM–receptor interaction," and "HALLMARK: angiogenesis" (Fig. 3a and Supplementary Fig. 9c).

To identify more robust predictive tumor expression markers associated with responsiveness to 5-FU and oxaliplatin-based

**Fig. 2 Comprehensive genomic profiles associated with responsiveness to 5-FU and oxaliplatin-based chemotherapy. a** Overall mutation patterns for both the responder (R) and non-responder (NR) groups. Tumor mutation burden (TMB) and clinicopathologic information (top), somatic mutation patterns of GC-associated genes (middle), and exclusively mutated cancer-associated genes (mutation frequency >20% only in the R or NR groups; bottom) were compared between the R and NR groups. **b** Trinucleotide mutation frequencies of the R and NR groups (top) and mutational signatures of single base substitutions (middle). Each mutational signature was grouped by the proposed etiology. Representative mutational signatures of 13 responders and 11 non-responders are shown with violin plots (bottom). In the box and whisker plots, data are presented as median value and standard deviation (SD), and the bottom and top edges of the box indicate the 25th and 75th percentiles, respectively. **c** Somatic indel frequencies of the R and NR groups (top) and differential mutational signatures of small insertions and deletions (bottom). In the box and whisker plots, indel signature proportions of 13 responders and 11 non-responders are presented as median value and SD, and the bottom and top edges of the box indicate the 25th and 75th percentiles, respectively. **d** Gene-level somatic copy number alterations in GC-associated genes (top) and differentially altered cancer-associated genes between the R and NR groups (bottom; Two-sided Fisher's exact test $P < 0.1$. Data are provided in Source data).

chemotherapy, we also analyzed the transcriptomes of matched human cancer tissues, and DEGs that were regulated in both patient and PDX tumors were compared between groups. A total of 40 upregulated genes shared between patient and PDX tumors were identified for the R group (log$_2$ fold-change $\geq 1.5$, $P < 0.1$), including *MDM2*, *TMPRSS2*, and *POUR5F1* (Fig. 3b). In contrast, 83 genes, including *FGFR1*, *PDGFRA*, *MLLT11*, and *COL1A1*, were upregulated in the NR group for both patient and PDX tumors (log$_2$ fold-change $\leq -1.5$, $P < 0.1$; Fig. 3b). A complete list of 123 DEGs is shown in Supplementary Data 2. Data integration between both exome and transcriptome analyses revealed fewer *TP53* mutations and increased p53 activity, based on increased p53 target gene (*MDM2* and *CDKN1A*) expression[17,18], in the R group (Fig. 3b), suggesting that intact p53 signaling likely enhances responsiveness to 5-FU and oxaliplatin-based chemotherapy. Protein–protein interaction (PPI) networks and the functional enrichment analysis of the 83 commonly upregulated genes between patient and PDX tumors in the NR group showed the significant functional enrichment of "ECM–receptor interaction," "Focal adhesion," and "PI3K-Akt signaling pathway" gene sets (Fig. 3c). Of the 83 upregulated gene in the NR group, 19 genes were previously reported to be upregulated in EMT[19] and 23 genes were associated with mesenchymal phenotype of GC[20], and many of these genes were subtype marker genes in previous studies (Supplementary Fig. 10a). In addition, EMT marker genes such as *SNAI1*, *MMP9*, and *FN1* showed significantly higher expression in the NR group (Supplementary Fig. 10b). These data emphasize the importance of p53 signaling and ECM interaction pathways for resistance to 5-FU and oxaliplatin-based chemotherapy.

**Tumor microenvironment (TME) reflects cancer cell characteristics based on responsiveness to 5-FU and oxaliplatin-based chemotherapy.** Because ECM–receptor interaction pathway activation was identified in the NR group of our PDX cohort (Fig. 3a, c), we further investigated the TME composition. In PDX models, TME components, including immune and stromal cells and non-cellular matrix proteins, are usually replaced by cells and matrix components of mouse origin[21,22]. Therefore, we analyzed mouse-originating RNA reads from the WTS analysis as representative of the TME[23].

GSEA analysis of the TME, using mouse-originating sequenced reads from PDX models, revealed the highly enhanced metabolic features of the R group, including "Fatty acid metabolism" and "Xenobiotic metabolism" (Fig. 4a and Supplementary Fig. 11a), consistent with the patterns observed for human cancer cells. In addition, gene sets associated with cell-to-cell and cell-to-ECM interactions, such as "Transforming growth factor-β (TGF-β) signaling," "Angiogenesis," and "Wnt/β-Catenin signaling," were enriched in the NR group (Fig. 4a and Supplementary Fig. 11a), consistent with the GSEA results from human cancer cells. Several TGF-β signaling pathway genes, such as *Eng*[24] and *Jag2*[25],

were upregulated in the NR group tumors (Fig. 4b), and *Eng* and *Jag2* genes were also reported to be involved in the regulation of endothelial cell migration and endothelium development[26,27]. Comparing TME DEGs between groups for both patient (from the whole transcriptome) and PDX models (from mouse reads), we identified 27 and 23 genes enriched in the R and NR groups, respectively (Fig. 4c). Network analysis, using the STRING database for 23 TME-associated genes in the NR group, showed the functional enrichment of genes regulating angiogenesis, including *Cdh5*, *Tie1*, and *Dll4* (Fig. 4d). These data suggested that the upregulation of metabolism-related pathways, in both tumor cells and the TME, cooperate to induce a favorable response to 5-FU and oxaliplatin-based chemotherapy, whereas cell-to-cell or cell-to-ECM interactions and angiogenesis were associated with an unfavorable response to 5-FU and oxaliplatin-based chemotherapy.

We also investigated the TME cellular composition, using transcriptome data for both patient and PDX tumors. The stromal scores of patient tumors were slightly higher in the NR group than in the R group ($P = 0.09$), and several cell types, such as endothelial cells (Mann–Whitney $U$ test, $P = 0.09$), interstitial dendritic cells (Mann–Whitney $U$ test, $P = 0.09$), and fibroblasts (Mann–Whitney $U$ test, $P = 0.14$), were enriched in patient tumors from the NR group (Fig. 4e and Supplementary Fig. 11b). The enrichment of endothelial cells in patient tumors during cellular composition analyses was compatible with the increased expression of angiogenesis-related gene sets observed in the TME transcriptome analyses (Fig. 4d). The estimated proportion of endothelial cells in the PDX TME were higher in the NR group than in the R group, although this difference was not significant (Fig. 4e and Supplementary Fig. 11c). In addition, when we compared marker gene expression for TME components (leukocytes and cancer-associated fibroblasts) among PDX tumors, genes associated with cancer-associated fibroblasts were expressed at higher levels in the NR group than in the R group (Supplementary Fig. 11d). These data suggested that stromal cells in the TME, such as endothelial cells and fibroblasts, and their interactions with cancer cells play important roles in responsiveness to 5-FU and oxaliplatin-based chemotherapy.

**Developing a predictive model of responsiveness to 5-FU and oxaliplatin-based chemotherapy in GC.** Since existing molecular subtyping methods based on hundreds of genes are less effective, we developed a predictive model of responsiveness to 5-FU and oxaliplatin-based chemotherapy with the fewest possible markers. After developing a predictive model based on 123 DEGs using Bayesian Compound Covariate Predictor (BCCP) algorithm, we reduced to 30 markers with high gene weight to create a simplified model with a correlation coefficient of 0.908 (with cross-validation $P$ value < 0.01, sensitivity for Rs of 0.86 and specificity of 0.89) (Supplementary Figs. 12 and 13). The high-weighted genes of the 30-gene prediction model included genes related to

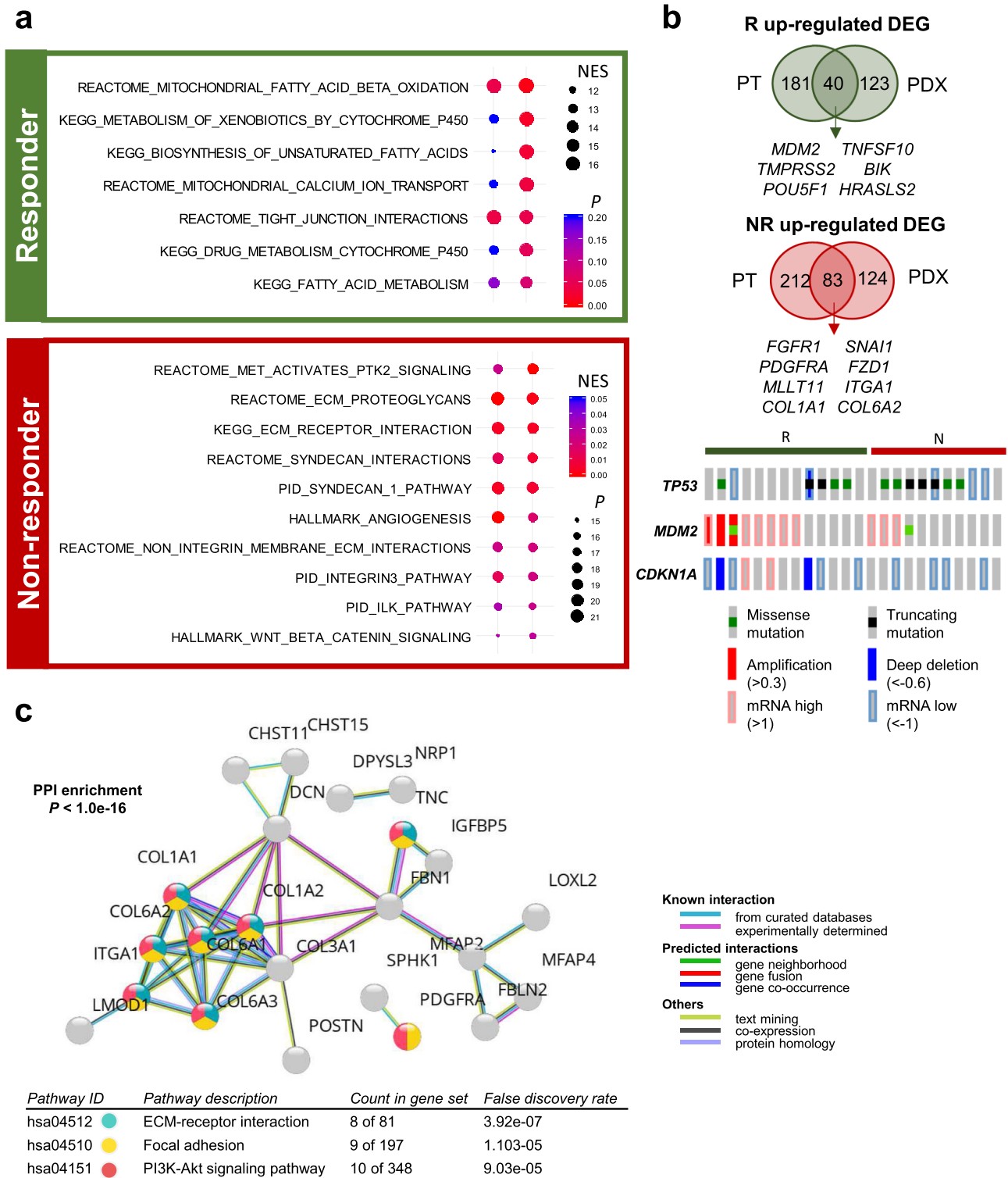

**Fig. 3 Identification of response-associated gene expression features for 5-FU and oxaliplatin-based chemotherapy. a** Gene set enrichment analysis (GSEA) of the R versus NR PDX models, using RNA sequencing data. GSEA demonstrated that metabolism-related pathways were enriched in the R group and Wnt/β-catenin and angiogenesis signaling pathways were enriched in the NR group. Normalized enrichment score (NES) and nominal *P* value is described. **b** Identification of core differentially expressed genes (DEGs) between the R and NR groups in both patient tumors and PDX tumors (Top). A total of 40 upregulated genes were identified in the R group and 83 upregulated genes were identified in the NR group. The bottom panel shows the genetic and transcriptomic alterations in *TP53*-associated genes in the R and NR groups. **c** Protein–protein interaction enrichment analysis based on STRING network for the 83 DEGs that are upregulated in the NR group. Enrichment *P* value are corrected for multiple testing using the method of Benjamini and Hochberg.

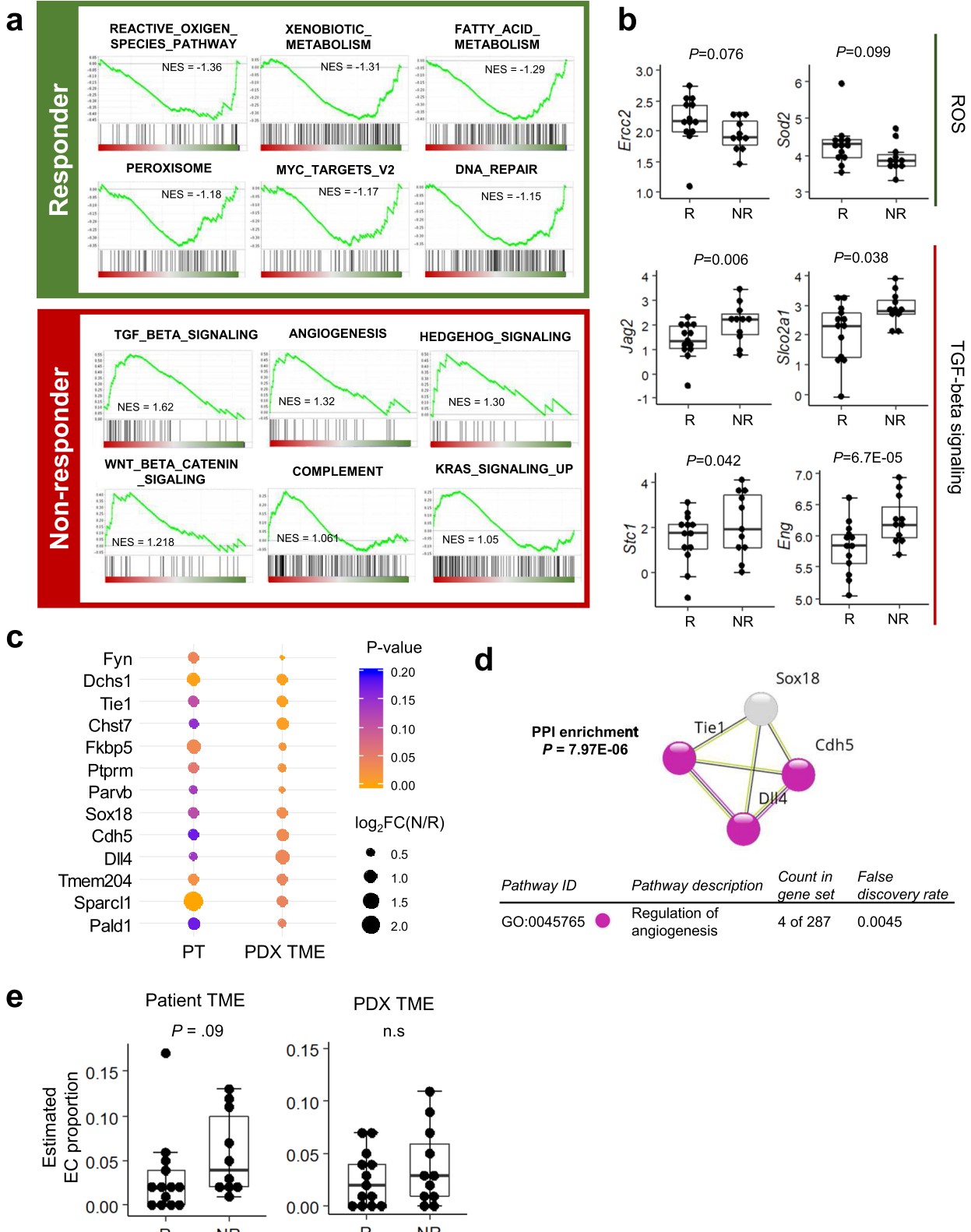

apoptosis, metabolism, ECM interactions, and EMT (Fig. 5a). The predictive model was applied to other independent GC cohorts to confirm the predictive value of this model by comparing the estimated responsiveness with patients' outcomes. Patients in cohort of ACRG[2], TCGA[6] and Singapore-Duke study[28] did not receive neoadjuvant therapy and some patients received adjuvant therapy with 5-FU-based regimens. Applied to 300 GC patients of

ACRG, 165 and 113 patients were assigned to R-like and NR-like, respectively (Supplementary Table 5). Survival analysis showed that NR-like patients had worse prognosis than R-like patients in both relapse-free ($P < 0.0001$ for all patients and $P = 0.061$ for patients with adjuvant therapy by log-rank test) and overall survival ($P < 0.0001$) (Fig. 5b and Supplementary Fig. 14a–c). In 262 TCGA GC patients, patients assigned to R-like ($n = 105$)

**Fig. 4 Expression profiles for the tumor microenvironment (TME) that are associated with responsiveness to 5-FU and oxaliplatin-based chemotherapy. a** GSEA of the presumed TME expression pattern, based on mouse sequencing reads from PDX tumors. The responder (R) group shows the enrichment of reactive oxygen species (ROS) and metabolism pathways, and the non-responder (NR) group shows the enrichment of TGF-β, Wnt/β-catenin, and angiogenesis signaling pathways. **b** Core gene expression from the top gene sets identified in the GSEA results. Log$_2$FPKM expression levels are shown with box plots ($n = 13$ for Rs, $n = 9$ for NRs). Data are presented as median value and standard deviation (SD), and the bottom and top edges of the box indicate the 25th and 75th percentiles, respectively. The two-tailed Wald test $P$ values are shown. **c** Candidate TME genes were selected by comparing patient tumors and PDX TME expression levels between the R and NR groups. **d** Protein–protein interaction (PPI) enrichment analysis for 23 candidate TME genes in the NR group, showing the protein–protein interactions associated with the regulation of angiogenesis. Enrichment $P$ values are corrected for multiple testing using the method of Benjamini and Hochberg. **e** Estimated endothelial cell (EC) proportion in both patient and PDX tumors ($n = 13$ for Rs, $n = 9$ for NRs). Data are presented as box and whisker plot with median value and SD, and the bottom and top edges of the box indicate the 25th and 75th percentiles, respectively. Two-tailed Mann–Whitney test $P$ values are shown. n.s. not significant.

showed significantly better prognosis in both relapse-free and overall survival than patients assigned to NR-like ($n = 84$) (Fig. 5c, Supplementary Table 5, and Supplementary Fig. 15). In 200 GC patients of Singapore-Duke study, patients were classified to 105 R-like and 79 NR-like subgroups (Supplementary Table 5), and NR-like patients had worse prognosis than R-like patients in overall survival ($P = 0.0013$ by log-rank test; Fig. 5d and Supplementary Fig. 16). When compared with subtypes in each cohort, most MSI-type samples in the ACRG and TCGA cohorts were classified into R-like groups, and most EMT-type sample in the ACRG cohort and most invasive samples in Singapore-Duke study were assigned to NR-like group (Fig. 5e–g). Taken together, we suggest that 30-gene-based classifier model has a predictive value for FOLFOX responsiveness and GC patient prognosis.

## Discussion

In this study, we comprehensively investigated genomic and transcriptomic differences between Rs and NRs to 5-FU and oxaliplatin-based chemotherapy, estimated from PDX experiments, and identified several important tumor biology pathways associated with drug response (Fig. 6a, b): (1) defective p53 signaling, in NRs; (2) increased metabolic processes, in Rs; and (3) increased cell-to-cell and cell-to-ECM interactions, in NRs. Defects in the p53 pathway have been suggested as determining factors for the sensitivity of 5-FU-based chemotherapy, in several cancer types[12,14,15], because p53 status is strongly associated with apoptosis and tumor-suppressing effects. In addition, several genes were reported to play roles in the resistance to 5-FU-based chemotherapy by regulating p53-dependent pathway[29,30]. Our genomic and transcriptomic analyses demonstrated a lower mutation frequency for *TP53* and the increased expression of p53 target genes, such as *MDM2* and *CDKN1A*[17,18], in the R group (Fig. 3b and Supplementary Fig. 7c). NRs with no *TP53* mutation also showed low p53 activity (group average: −0.13; Supplementary Fig. 7c), suggesting that p53 function is inactive or defective in most NRs. Furthermore, among 343 genes, directly regulated by p53[31], several genes including *SFN* (associated with EMT; $P = 0.0092$), *FDXR* (associated with metabolism; $P = 0.0107$), and *BBC3* (associated with apoptosis; $P = 0.0349$) were significantly downregulated in the NR group. These data suggest that defective p53 signaling is one of the predictive features for 5-FU and oxaliplatin-based chemotherapy resistance.

Our transcriptomic analyses demonstrated metabolism-related gene set enrichment in the R group and the activation of cell-to-cell and cell-to-ECM interactions in the NR group, which were reproducibly detected in analyses of both cancer cells and the TME (Figs. 3a and 4a), indicating the importance of interactions between cancer and stromal cells in the TME for the development of resistance to 5-FU and oxaliplatin-based chemotherapy. Several soluble factors in the TME appear to play critical roles in determining chemotherapy responsiveness. TGF-β plays an important role in drug resistance against both targeted and

conventional agents. Recent studies showed that the treatment of triple-negative breast cancer xenografts with paclitaxel induced autocrine TGF-β signaling, cancer stem cell formation, and drug resistance[32]. Clinical trials are ongoing for the combined treatment of chemotherapeutic drugs and the TGF-β receptor I inhibitor LY2157299, in glioblastoma (NCT01582269), hepatocellular carcinoma (NCT01246986), and pancreatic cancer (NCT01373164). Wnt/β-catenin signaling is reportedly involved in a multitude of developmental processes, the regulation of cell proliferation, differentiation, migration, genetic stability, and apoptosis[33]. Although therapeutic agents that specifically target the Wnt pathway have recently entered clinical trials, the detailed mechanisms through which the Wnt pathway generates chemotherapeutic drug resistance remain unclear[33]. TGF-β signaling and Wnt/β-catenin were highly associated with the EMT process[34,35]. Previous studies have suggested that TGF-β-induced EMT confer GC cells resistance to 5-FU[36,37], and suppression of EMT via diverse approaches increased the sensitivity to 5-FU-based chemotherapy in GC[38,39]. The role played by the EMT in reduced therapeutic efficacy is evident in several tumors[40–42]. The 5-FU-resistant cell lines showed typical EMT alterations, including increased invasiveness, the upregulation of mesenchymal markers, the increased expression of EMT-related proteins, and the downregulation of epithelial markers[43]. Our transcriptomic data showed that several EMT-related genes were significantly upregulated in the NR group (Supplementary Fig. 10a, b), and in our prediction, ACRG EMT subgroup and Singapore-Duke invasive subtypes were mostly assigned to NR-like (Fig. 5e, g). These data emphasized that cancer cells and the TME work cooperatively during resistance to 5-FU and oxaliplatin-based chemotherapy, by activating comparable pathways.

The effect of defective MMR pathway in 5-FU-based chemotherapy responsiveness is quite controversial[11,44]. In previous clinical trials, the CLASSIC and the MAGIC trials demonstrated that patients with MSI high (MSI-H) GC showed no survival benefit and worse survival from 5-FU-based adjuvant chemotherapy, respectively[45,46]. However, recent study from 162 Korean patients with MSI-H GC demonstrated that the group with 5-FU-based adjuvant chemotherapy showed better overall and disease-free survival compared to the group with no adjuvant chemotherapy[47], suggesting a possible role of a defective MMR pathway in the responsiveness of 5-FU-based adjuvant chemotherapy. In our data, despite a higher frequency of MSI-H (Rs: 38.5%, NRs: 27.3%) and following higher frequencies of defective MMR mutational signatures in Rs (Fig. 2b, c), those features are not general characteristics of the R group. Recent studies reported molecular subtyping of MSI-H tumors in colorectal and GCs and revealed prognostic difference among MSI-H tumors[48,49], suggesting that patients with MSI-H are heterogeneous in terms of tumor characteristics and prognosis. Therefore, the effect of a defective MMR pathway on the responsiveness of 5-FU-based

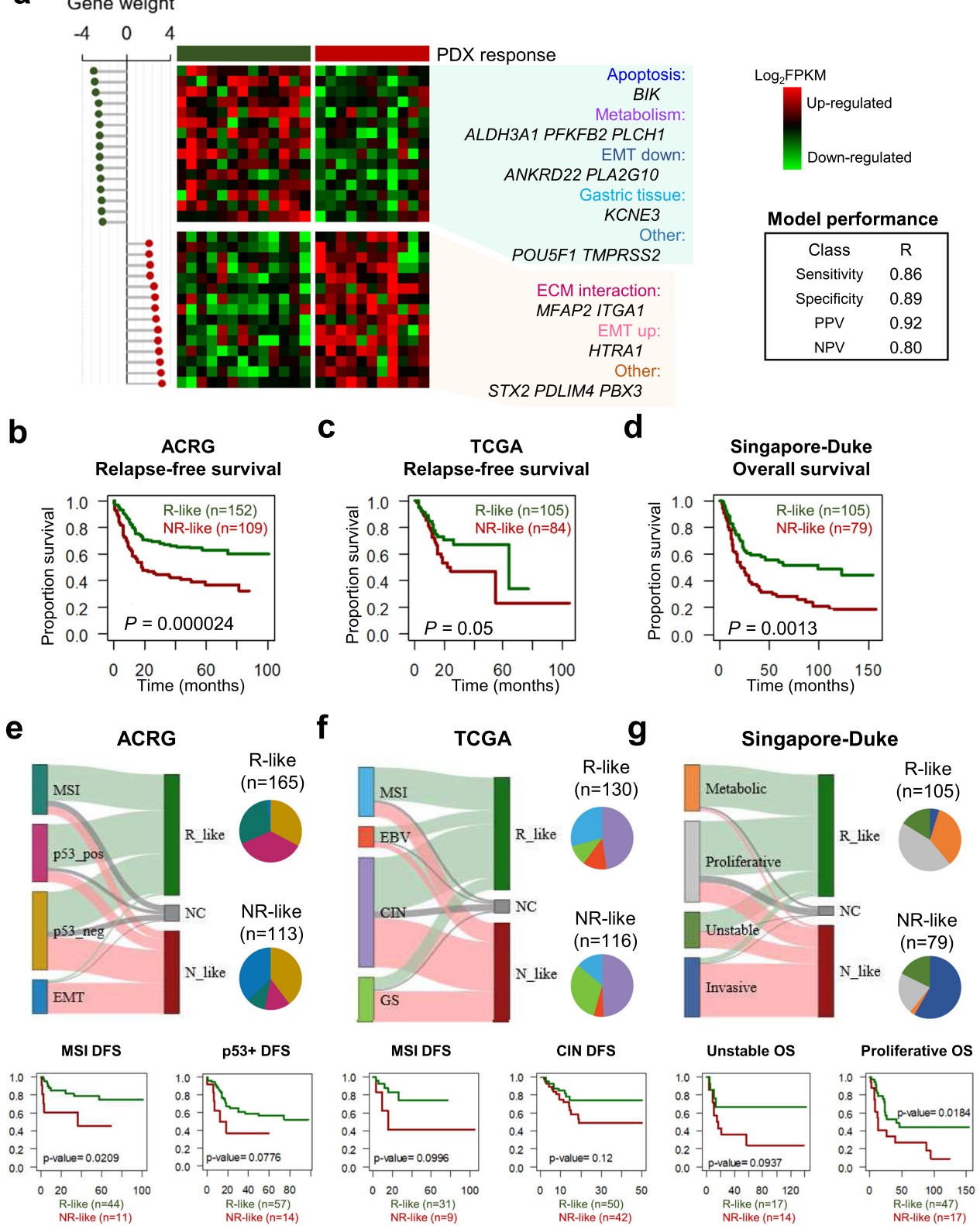

**Fig. 5 Development of a response prediction model for 5-FU-based chemotherapy. a** The gene expression of 30 classifier genes from among the core DEGs in PDX tumors. **b–d** Survival analyses for GC patients in ACRG (**b**), TCGA (**c**), and Singapore-Duke (**d**) studies, according to prediction model. Patients were stratified by our prediction model, and the responder-like (R-like) patient group showed a significantly better prognosis than the non-responder-like (NR-like) group. *P* values were calculated from the log-rank test. ACRG the Asian Cancer Research Group. **e–g** Reassignment of GC patients from the ACRG study (**e**), TCGA (**f**), and Singapore-Duke (**g**) studies according to prediction model (Top). Survival analyses for subgroups in ACRG (**e**), TCGA (**f**), and Singapore-Duke (**g**) studies, according to prediction model (bottom). *P* values were calculated from the two-sided log-rank test.

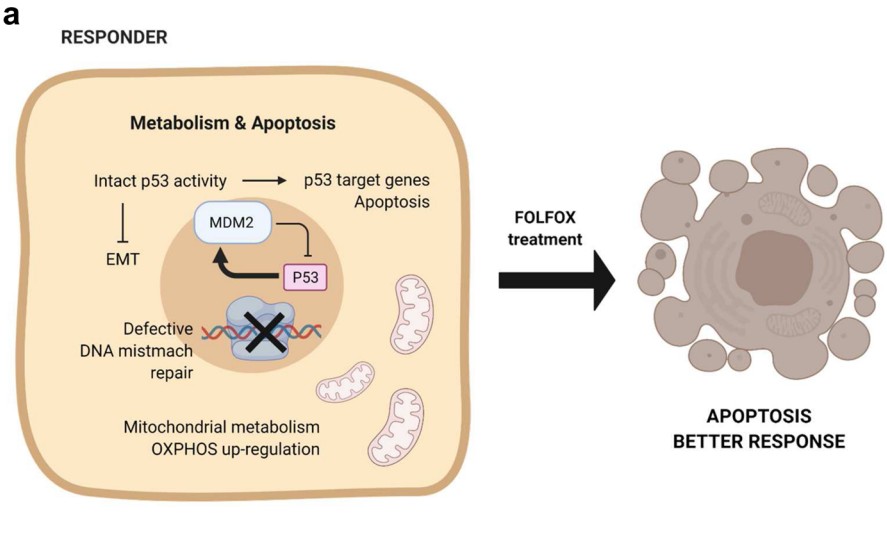

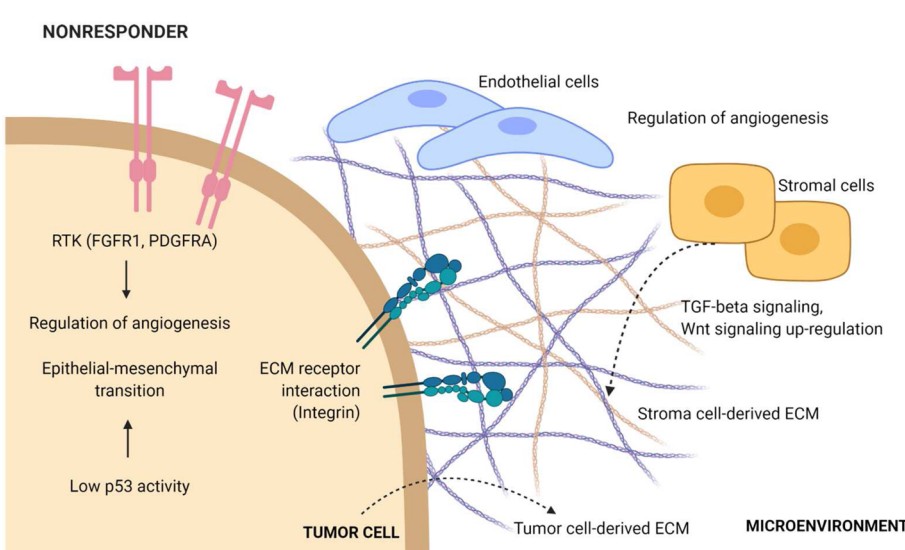

**Fig. 6 Summary of integrative characteristics associated with 5-FU-based therapy responding and non-responding tumors. a** Representative pathways enriched in the R group. **b** Representative pathways enriched in the NR group. The figures were generated with BioRender.

adjuvant chemotherapy needs to be evaluated by randomized prospective clinical trials with a large cohort of MSI-H GC patients.

The responsiveness prediction model, built using DEGs as biomarkers in a machine learning-based prediction model and 30 gene expression signatures, correctly classified our PDX models and GC patient cohorts. Several efforts have been made to identify GC biomarkers, including a predictive test for adjuvant chemotherapy response, which compared the expression levels of four classifier genes in formalin-fixed, paraffin-embedded tumors between patients who underwent surgery alone and those who received postoperative chemotherapy[50]. However, this study was unable to completely separate the effects of surgery from the effects of chemotherapy. Recently, to identify predictive biomarkers for 5-FU + oxaliplatin-based neoadjuvant chemotherapy, Li et al. compared the sequencing data profiles of biopsy samples collected from 35 Chinese GC patients who underwent neo-adjuvant therapy[51]. By comparing the multi-omics characteristics of 17 Rs and 18 NRs, they demonstrated that MSI and *MDM2* amplification were observed in non-responsive tumors, whereas *MYC* amplification was observed in responsive tumors.

Some discrepancies exist between this previous study and our data, especially for the effects of MSI in drug responsiveness. Although our observations were cross-validated in both patient and PDX tumors, detailed explanations regarding these differences must be further evaluated.

PDXs are increasingly used as drug development tools in preclinical settings and have been shown to recapitulate the histology and behavior of the cancers from which they are derived. However, whether PDXs mimic the clinical outcomes of patients remains controversial. The Golub group tracked the dynamics of copy number alterations during passaging in 1110 PDX models from 24 types of cancers, collecting the relevant genetic data[52]. The results revealed variations in the mouse data compared with the genetic data from human tumor cells after the tumor tissues were transplanted into mice. The researchers suggested that these changes may cause different responses to cancer drugs in the PDX models. In contrast, the Sidransky group established PDX models of 92 patients with various solid cancers and conducted a dosing study, which identified correlations as high as 87% between the response to drugs in patients and the associated PDX models[53]. Also, Wong et al. performed a retrospective population

pharmacokinetic-pharmacodynamic analysis of relevant xenograft efficacy data for eight anticancer drugs with known clinical outcome, including 5-FU. This group suggested that agents that can achieve 60% TGI or higher in preclinical models are more likely to drive clinical efficacy in man[9]. Our data demonstrated that the estimated responsiveness to 5-FU and oxaliplatin-based chemotherapy in PDX models was highly correlated with clinical outcomes in the adjuvant setting (Fig. 1c–e) and that the prediction model, developed from analyses of PDX tumors, was able to predict patients' drug responsiveness and prognosis (Fig. 5). There are some limitations in analyzing human samples compared to PDX models. It is difficult to obtain large quantity of cancer tissues from patients, and tumor cell purity in tissues are more diverse in human samples than in PDX tumors. Patients' general conditions and environmental factors are hard to normalize in human study, which requires large size of patient cohort. Therefore, through multiple PDXs, integration of drug responsiveness, and genomic/transcriptomic profiling will provide additional prediction model for treatment responsiveness other than 5-FU and oxaliplatin-based chemotherapy.

## Methods

**GC patient sample collection**. This study was approved by the institutional review board of the Seoul National University Hospital (No. C-1402-054-555), in accordance with the Declaration of Helsinki. All samples were obtained with informed consent at the Seoul National University Hospital. Tissue samples of GCs, paired normal gastric tissues, and blood samples were obtained from individuals who underwent gastrectomy at Seoul National University Hospital between 2014 and 2017. Patients who were treated preoperatively with chemotherapy or chemoradiation therapy were excluded from this study. Immediately after sample acquisition, tumor samples were transferred to RPMI 1640 medium (Thermo Fisher Scientific), containing 1% penicillin/streptomycin (Thermo Fisher Scientific), for PDX generation.

**Generation of PDX models**. Mice were cared for according to the institutional guidelines of the Institutional Animal Care and Use Committee of Seoul National University Hospital (No. 14-0016-C0A0). For PDX models, surgically resected tissues were minced into pieces, approximately 2 mm in size, and injected into the subcutaneous area of the flanks of 6-week-old NOD/SCID/IL-2γ-receptor null (NSG mice, The Jackson Laboratory) female mice. The tumor volumes and body weights of the mice were checked once or twice, weekly. Tumor volume was calculated as (length × width$^2$)/2. Tumor formations in the implanted site >500 mm$^3$ in size were considered successful engraftments. Mice with successful engraftment were sacrificed, and tumor tissues were excised and stored. For each tumor, tumor tissues were divided and stored for three purposes: (1) tumor tissues were cryopreserved in liquid nitrogen, for use in the next passage of PDXs; (2) tumor tissues were frozen in liquid nitrogen, for genomic analyses; and (3) tumor tissues were fixed in 10% formalin solution, for histological analyses.

**In vivo pharmacological studies**. In PDX mice bearing subcutaneous engraftments of previously established PDX tumors, drug treatments began after the tumors reached approximately 200 mm$^3$. Mice were divided randomly, into control and 5-FU + oxaliplatin-treated groups, with five mice in each group. 5-FU (Selleckchem, 5 mg/kg, weekly) and oxaliplatin (Selleckchem, 50 mg/kg, weekly), in saline, was administered via intraperitoneal injection for 21 days. Tumors volumes were checked three times, weekly, calculated as (length × width$^2$)/2.

To evaluate the responsiveness, we combined two criteria comparing relative tumor volume changes between drug- and vehicle-treated arms: (1) drug treatment significantly inhibited tumor growth [two-way ANOVA $P < 0.0001$ for the R group, $P > 0.05$ for the NR group]; and (2) the TGI (%) was >60% for the R group (Supplementary Table 2). TGI is calculated via the following equation:

$$\text{TGI (\%)} = \frac{\text{TV}_{\text{vehicle}} - \text{TV}_{\text{treatment}}}{\text{TV}_{\text{vehicle}} - \text{TV}_{\text{initial}}} \times 100 \qquad (1)$$

where $\text{TV}_{\text{vehicle}}$ is the tumor volume for the vehicle-treated animals at a specified end point time, $\text{TV}_{\text{initial}}$ is the initial tumor volume at the start of the treatment, and $\text{TV}_{\text{treatment}}$ is the tumor volume of the drug treatment groups at a specified end point time[9]. The intermediate group consist of samples that did not meet these criteria. The ANOVA test was performed using the SPSS software version 22 (IBM Corp., Version 22.0).

**Statistical analysis of clinical characteristics**. Clinical and pathological information regarding enrolled patients was collected prospectively. The following variables were analyzed to identify factors associated with responsiveness to 5-FU

and oxaliplatin-based chemotherapy: (1) the patient's age and sex, and (2) tumor pathological characteristics, including and pTNM stage, according to the Seventh edition of the Union for International Cancer Control, Lauren classification. Continuous variables are presented as the median and analyzed by Mann–Whitney $U$ test, using the SPSS software. Categorical variables are presented as the number and percentage and analyzed by Fisher's exact test. A $P$ value of <0.05 was considered significant.

**Isolation of nucleic acids and next-generation sequencing**. Genomic DNA was extracted from patient blood samples using the Gentra Puregene Blood Kit (Qiagen), and tumor DNA was obtained from patients and PDX tumors using the DNeasy Blood & Tissue Kit (Qiagen). Then 250 ng of DNA was sonicated with a Covaris S220 Focused-ultrasonicator, and 101-bp paired-end libraries were constructed with the SureSelect All Exon V5 Kit (Agilent). WES was performed on Illumina HiSeq 2000 instruments, with read lengths of 2 × 101 bp.

RNA extraction from non-tumor tissues and patient and PDX tumors was performed using TRIzol™ (Invitrogen). Samples with an RNA integrity number >5 were further processed. The 101-bp paired-end libraries were constructed with the TruSeq RNA Sample Prep Kit v2 (Illumina), using 1 µg of RNA. Whole-transcriptome sequencing (WTS) was performed on Illumina HiSeq 2000 instruments.

**Sequencing data processing using combined reference**. We built a combined reference genome of the human (GRCh37) and mouse reference genomes (GRCm38) and raw FASTQ files from WES and WTS were aligned with the combined reference genome. Sequenced reads from WES and WTS were aligned using a Burrows–Wheeler Aligner mem[54] and STAR aligner[55], respectively. The sorting and marking of duplicates were performed by Picard tools. After processing the binary aligned and mapping (BAM) files, mouse genome-aligned reads were removed for analyses of human cancer cells.

**Identification of somatic single-nucleotide variants and insertions/deletions (indels)**. Indel realignment and base recalibration of the BAM files were performed using the Genome Analysis Tool Kit (GATK) version 3.2. Somatic mutations were called using MuTect[56] and Indelocator[57] and annotated with Annovar[58]. Variants with total read depths <7 and alternate allele depth <4 were excluded. To select rare functional mutations, coding sequence mutations, with a population frequency of <0.01, were included, based on the ESP6500 and the total and East Asian populations of 1000 Genomes Project phase 3 and ExAC studies. SNPs with ≥1% minor allele frequency, based on the dbSNP flag, and error-prone variants on the segmentally duplicated region were removed.

Cancer-related genes were annotated using cancer gene databases, including TARGET (Tumor Alterations Relevant for GEnomics-driven Therapy; http://www.broadinstitute.org/cancer/cga/target), Cancer Gene Census[59], and known cancer genes from Vogelstein et al.[60]. Oncogenes and oncogenic/likely oncogenic mutations were annotated by OncoKB (http://www.oncokb.org).

**Decomposition of mutational signatures**. We estimated the amount of exposure to mutational processes for each tumor sample using MuTect results, based on the mutational signatures of the COSMIC database (http://cancer.sanger.ac.uk/cosmic/signatures). The R package deconstructSigs[61] was used to compute the mutational signatures, and tumor samples with matched germline data that harbored at least 20 mutations were included for the analysis.

**Analysis of clonal architecture**. To estimate tumor heterogeneity and changes in the clonal architecture of each pair of patient and PDX tumors, PyClone, a Bayesian clustering method relying on Markov Chain Monte Carlo, was applied[8]. Based on the allele frequencies of somatic mutations and copy number values from WES, PyClone clustered mutations that shifted together across tumors and predicted the cellular prevalence of each cluster. A beta-binomial emission was applied. Two pairs of patient and PDX tumors were excluded from this analysis due to the absence of a matched patient tumor and few mutual mutations identified between the patient and PDX tumors.

**Identification of SCNAs**. To identify SCNAs between Rs and NRs that may affect drug response, Z-transformed RPKMs (Reads Per Kilobase per Million mapped reads) of each tumor were calculated using CoNIFER[62], log normalized with matched germline data, and further segmented using DNA copy package (https://www.bioconductor.org/packages/release/bioc/html/DNAcopy.html). For tumor samples without matched blood, pooled blood data were used as a baseline. Amplification was defined as a value >0.2, whereas deletion was defined as a value <−0.2.

**Gene expression analysis**. The numbers of expressed reads for each gene were quantified using the HTSeq-count program, based on the GRCh37 Ensemble v65. Fragments per kilobase of exon per million mapped reads (FPKM) values were calculated and normalized with edgeR[63]. Of the 62,000 Ensemble genes, we first

removed genes with median FPKM values < 1. GSEA between Rs and NRs was performed on GenePattern module[64] using the Hallmark gene sets.

The extraction of DEGs was performed using DESeq2 package[65], which uses raw counts. After DEG extraction, we applied further filtration steps as follows: adjusted $P$ value < 0.1, |log$_2$ fold-change| ≥ 1.5. Using filtered DEGs, we performed pathway analysis using the Kyoto Encyclopedia of Genes and Genomes (KEGG) analysis tool and Gene Ontology database and determined statistical significance with The Database for Annotation, Visualization and Integrated Discovery functional annotation tool. Clustering was performed based on log-transformed and gene-centered FPKM values, using the cluster3.0 program, and visualized with Treeview. Enrichment of PPIs among DEGs was analyzed based on KEGG, using the STRING database.

**TME analysis**. During the xenotransplantation, human stromal cells are replaced with mouse-derived stromal cells, and PDX tumors composed of human tumor cells become surrounded by murine stroma. Thus, tumor-specific gene expression was observed through human genome-aligned reads, and stroma-specific gene expression could be separately analyzed based on mouse genome-aligned reads[66]. Murine gene expression was evaluated and counted based on the *Mus Musculus* GRCm38 Ensembl v78 database, which was converted into FPKM using edgeR. DEG analyses were performed using DESeq2 and genes with |log$_2$ fold-change| > 1.5 and adjusted $P$ value < 0.05 were evaluated further. GSEA and PPI analysis were performed, as in the tumor analyses.

xCell estimates the abundances of 64 stromal and immune cell types[67]. We converted mouse genes to orthologous human genes, and xCell was applied to determine the microenvironment components of PDX tumors compared with those in patient tumors.

**Development of the response prediction model to 5-FU and oxaliplatin-based chemotherapy**. To develop a predictive classification model for 5-FU-based chemotherapy responsiveness, 30 signature classifier genes were selected from core 123 DEGs of tumor analysis. Using these classifiers, we developed a model using the BCCP algorithm assuming a Gaussian distribution, of BRB-ArrayTools[68], which was cross-validated by the leave-one-out cross-validation. Robustness of the trained model was evaluated by the Receiver Operating Characteristics curve. We applied this prediction model to TCGA, ACRG (GSE62254)[2], and Singapore-Duke (GSE15459)[28] cohorts and reclassified tumors into R-like and NR-like groups. Clinicopathologic analysis was performed between the R- and NR-like groups. Survival analysis was conducted using the Kaplan–Meier method and log-rank test. The Sankey diagram was built using R library networkD3.

**Reporting summary**. Further information on research design is available in the Nature Research Reporting Summary linked to this article.

## Data availability

The PDX sequencing data generated in this study have been deposited in the European Nucleotide Archive (ENA) repository under accession code PRJEB40936. The TCGA validation data used in this study are available in the FIREHOSE database [http://firebrowse.org/?cohort=STAD]. The ACRG expression data used in this study are available in the Gene Expression Omnibus (GEO) database under accession code GSE62254. The Singapore-Duke expression data used in this study are available in the GEO database under accession code GSE15459. Source data are provided with this paper.

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

## Acknowledgements

This research was supported by the Korean Healthcare Technology R&D project through the Korean Health Industry Development Institute (KHIDI) funded by the Ministry of Health & Welfare, Republic of Korea (grant No. HI13C2148); Bio & Medical Technology Development Program of NRF, funded by the Ministry of Science & ICT (grant Nos. 2018M3A9F3056902 and 2019M3E5D4066900); the Seoul National University Hospital Research Fund (grant No. 2620190020); Basic Science Research Program through the National Research Foundation of Korea (NRF) funded by the Ministry of Education (2020R1A6A1A03047972); the operational funds from The First Affiliated Hospital of Xi'an Jiaotong Univeristy; and the National Cancer Institute of Health (Award Number P30CA0341906). C.L. was a distinguished Ewha Womans University Professor supported in part by the Ewha Womans University Research grant of 2016-2018.

## Author contributions

S.-A.I., H.-K.Y., C.L., and J.-I.K. conceived the study. D.N., J. Chae, S.-Y.C., A.M., S.-A.I., H.-K.Y., C.L., and J.-I.K. designed the study. A.M., Y.-J.K., K.-H.L., T.-Y.K., Y.-S.S., S.-H. K., H.-J.L., S.-A.I., and H.-K.Y. collected patient tumor samples. D.N., J. Chae, S.-Y.C., W.K., A.L., S.M., J.K. M.J.K., J. Choi., W.L., D.S., and A.M. performed analysis. D.N., J. Chae, S.-Y.C., A.M, K.-H.L., T.-Y.K., Y.-S.S., S.-H.K., H.-J.L., W.-H.K., H.P., S.-A.I., H.-K.Y., C.L., and J.-I.K. interpreted results. D.N., J. Chae, and S.-Y.C. wrote initial draft. D.N., J. Chae, S.-Y.C., A.M., S.-A.I., H.-K.Y., C.L., and J.-I.K. contributed to writing and editing the manuscript.

## Competing interests

The authors declare no competing interests.
