## [Peer Review File · Nature Communications]

Reviewers' Comments:

Reviewer #1:

Remarks to the Author:

The paper by Na et al. describes the identification of a 29-gene prediction model to determine the responsiveness to 5-FU and oxaliplatin-based chemotherapy in gastric cancer patients. The authors treated 31 gastric PDXs with a combination of 5-FU-oxaliplatin and compared their response to that of the patients from which PDXs have been derived. The genomic and transcriptomic analysis of responders and non responders allowed the identification of pathways and genes dysregulated in responders and non responders.

The identification of biomarkers of sensitivity/resistance to chemotherapy is clearly needed in the clinic to improve the efficacy of treatment. The work is thus potentially interesting.

Other authors have already provided multiomics analyses to investigate resistance to chemotherapy in gastric cancer with different results. Another critical problem is the lack of in depth analysis of the mechanisms through which the identified genes could play a role in sustaining resistance/response.

In the discussion the authors state that they have identified several important pathways associated with drug response. 1) defective DNA MMR pathway is observed in the response group. This is poorly in agreement with published data and clinical observation. Thus it should be further investigated; 2) defective p53 signaling in non responders. The presented data are not strong enough; 3) EMT alterations. Clear evidence of EMT presence should be provided in PDXs and in cell lines. By the way, a list of the used cell lines has not been provided; details of the experiments are missing as well as the partitioning of the cell lines between responders and non responders. This would not allow other researchers to confirm the data.

The way response has been evaluated in PDXs is questionable. To make a clinical comparison, response should be evaluated in relation to the volume at treatment start. This would allow to define progressive disease, stable disease and partial/complete response as it happens in patients.

Reviewer #2:

Remarks to the Author:

Predictive biomarkers for responsiveness to 5-fluorouracil and oxaliplatin-based chemotherapy in gastric cancers through the integrative profiling of patient-derived Xenografts

This study by Na and colleagues aims to identify predictive biomarkers for 5-FU/oxaliplatin (5-FU/oxa) in gastric cancer using patient derived xenografts (PDXs). Analyzing 31 PDXs and treated with 5-FU/oxa, they show that the PDX responses are consistent with the clinical outcomes of the original patients. Comparing 5-FU/oxa responders (R) and non-responders (NR), they identify at the DNA level certain genes mutated between both groups (eg BRD3 in R, NCOR2 in NR), and also at the copy number level (CCND1 in R, ETV1 in NR). The functional implications of these genomic differences remains to be determined. At the transcriptome level, they discover pathways related to cell-to-cell and cell-to-extracellular matrix interactions enriched among NR PDXs, which is further confirmed by examining integrative differentially expressed genes common to PDXs and matched tumors, and also mouse-specific reads in the PDX – collectively emphasizing a role for the TME in NRs. Developing a 29-gene 5-FU/oxa prediction model, they apply the model to three independent GC cohorts, confirming poor survival outcomes among cases classified as non responder-like.

Major Comments

1) Can the authors provide any data regarding genomic stability of the PDXs, in terms of driver gene profiles or clonality profiles over repeated passage?

2) In Fig. 1b, why are three NRs chosen amongst the 'intermediate' range of PDXs? Likewise, why is one 'intermediate' PDX intermixed among the Rs?

3) When analyzed across the entire PDX cohort, is there is an association between tumor mutation

burden, MMR mutation signatures, and the observed downregulation of MMR essential genes?

4) In Fig. 2A, 9 of the PDXs are designated as MSI-positive, which seems (to this Reviewer) rather high. The increase in TMB among these tumors relative to the non-MSI PDXs also does not seem that high. Can the authors confirm the MSI status of these, perhaps using the standard microsatellite panel of MLH1/MSH2 IHC?

5) In order to pinpoint differentially regulated genes between R and NR, the authors integrate transcriptomic data from both PDX and matched human tissues. This raises the question of whether the findings highlighted (for example, the enrichment of ECM-matrix in NR) might have been determined by analysis of human tumors alone, without relying on PDXs. Can the authors demonstrate or clarify the value of using the PDX system (also given that there is a correlation between patient response and PDX response).

6) It is not clear to me why, in formulating their 29-gene model, the authors choose to include MLH1. This seems quite arbitrary to me and is not consistent with the unbiased data-driven approach used to select the other genes.

7) As mentioned in the Discussion, several studies in gastric cancer have pointed to the existence of a 'mesenchymal' subtype of gastric cancer (Lei et al., 2013; ACRG study (MSS/EMT); Oh et al., 2018). To what extent does the author's 29 gene signature overlap with these in a multivariate analysis to provide independent predictive power?

8) Can the authors definitely address whether or not the 29-gene signature is truly predictive or also prognostic in nature? This could be addressed by testing it in a cohort of untreated gastric tumors.

Responses to Reviewer's comments:

Please find below our responses to each specific comment and suggestion raised by the two reviewers.

Reviewer #1:

1. In the discussion the authors state that they have identified several important pathways associated with drug response. 1) defective DNA MMR pathway is observed in the response group. This is poorly in agreement with published data and clinical observation. Thus it should be further investigated; 2) defective p53 signaling in non responders. The presented data are not strong enough; 3) EMT alterations. Clear evidence of EMT presence should be provided in PDXs and in cell lines.

: As the reviewer suggested, we added additional supporting evidence for the association of drug response with defective DNA MMR pathway, defective p53 signaling, and EMT alterations as below.

1) Defective DNA MMR pathway

We agree that the effects of a defective MMR pathway on the responsiveness of 5-FU-based adjuvant chemotherapy are controversial. In previous clinical trials, the CLASSIC and the MAGIC trials demonstrated that patients with MSI-H gastric cancer showed no survival benefit and worse survival from 5-FU-based adjuvant chemotherapy, respectively (Ann Surg. 2019, 270:309-16, PMID: 29727332; JAMA Oncol. 2017, 3:1197-203, PMID: 28241187). However, only 40 and 20 patients with MSI-H gastric cancer were included in the CLASSIC and the MAGIC trial, respectively. Recent study from 162 Korean patients with MSI-H gastric cancer demonstrated that group with 5-FU-based adjuvant chemotherapy showed better overall and disease-free survival compared to group with no adjuvant chemotherapy (Cancer Res Treat. 2020, 52:1178-1187, PMID: 32599979), suggesting a possible role of a defective MMR pathway in the responsiveness of 5-FU-based adjuvant chemotherapy. In our data, despite a higher frequency of MSI-H (R: 38.5%, N: 27.3%) and following enrichment of dMMR mutational signatures in responders, those features are not general characteristics of the responder group. Recent studies reported molecular subtyping of MSI-H tumors in colorectal and gastric cancers and revealed prognostic difference amongst MSI-H tumor (Cell Commun Signal. 2019, 17:79, PMID: 31331345; J Med Genet. 2021, 58:12-9, PMID: 32170001), suggesting that patients with MSI-H are heterogeneous in terms of tumor characteristics and prognosis. Therefore, the effect of a defective MMR pathway on the responsiveness of 5-FU-based adjuvant chemotherapy needs to be evaluated by randomized prospective clinical trials with a large cohort of MSI-H gastric cancer patients. As the reviewer suggested, we toned down the significance of defective DNA MMR pathway in drug responsiveness and removed this issue from main point of

our findings in the Discussion section. But we described this issue in the separate paragraph of Discussion section of the revised manuscript (p. 17).

2) Defective p53 signaling

Expression of p53 has been suggested as a predictor of the response to 5-FU-based chemotherapy in gastric cancer cell lines (Int J Oncol. 2004, 24(4):807-1, PMID: 15010816), and allelic imbalance of p53 was associated with resistance to 5-FU-based chemotherapy (Ann Surg Oncol. 2009, 16(10):2926-35, PMID: 15010816). And several genes were reported to play roles in the resistance to 5-FU-based chemotherapy by regulating p53-dependent pathway (Mol Carcinog. 2018, 57(6):722-734, PMID: 29436749; Comput Struct Biotechnol J. 2019, 30;18:125-136, PMID: 31969973; PLoS One. 2014, 9(2):e90155, PMID: 24587255).

In our data, non-responder showed as twice as a higher mutation frequency in *TP53* (64% in non-responder and 38% in responder), and p53 activity score estimated using representative p53 target genes *MDM2* and *CDKN1A* expressions were significantly lower in N (group average: -0.113) than R (0.193) as below. In addition, non-responders with no *TP53* mutation also showed low p53 activity (group average: -0.13) as below, suggesting that p53 function is inactive or defective in most non-responders.

Furthermore, among 343 genes, directly regulated by p53 (Oncogene 2017, 36:3943-56; PMID: 28288132), several genes were significantly down-regulated in NR group. Tumor suppressor protein 14-3-3 sigma (SFN), which was reported to be down-regulated in bladder cancer undergoing EMT (Mol Cell Proteomics 2004;3:410-9, PMID: 14736829), showed significant down-regulation in non-responders ($P = 0.0092$). Several other p53 direct target genes, such as *FDXR*, associated with metabolism ($P = 0.0107$), *BBC3* and *FAS* in intrinsic and extrinsic apoptotic signaling ($P = 0.0349$ and 0.0860 , respectively) showed significant higher expression in R group. These data suggest that defective p53 signaling is associated with the resistance to 5-FU-based chemotherapy in gastric

cancers. These comments are now included in the Result and Discussion section of the revised manuscript (p. 8, 15, 16) and Extended Data Figure 7c.

3) EMT alterations

Previous studies have suggested that TGFβ-induced EMT confer gastric cancer cells resistance to 5-FU (FASEB J. 2019, 33(5):6365-6377, PMID: 30802150; Cell Physiol Biochem. 2018, 47(4):1533-1545, PMID: 29940566), and suppression of EMT via diverse approaches increased the sensitivity to 5-FU-based chemotherapy in gastric cancer (Cancer Manag Res. 2018, 10:2729-2742, PMID: 30147370; Int J Biochem Cell Biol. 2018, 102:59-70, PMID: 29953965).

In our data, expression of representative EMT marker genes were significantly up-regulated in non-responder tumors as below. (asterisk: $P < 0.05$, cross: $P < 0.1$)

When we compared 123 differentially expressed genes (DEGs) between R and N in our study with ACRG study and Oh *et al.*, which suggested prognostic expression signature, total 43 of 123 genes were overlapped as below. Venn diagram was added in Extended Data Figure 10a. Several overlapping genes were associated EMT/mesenchymal types, and seven genes (*COL1A2*, *COL3A1*, *DKK3*, *ANTXR1*, *CMTM3*, *DPYSL3* and *HTRA1*) were common in both studies (Genes in blue indicate the genes associated with EMT). Several genes, associated with mesenchymal/EMT phenotype of gastric cancer from other reports, were significantly up-regulated in the non-responder group.

In addition, the p53-Mdm2 complex is known to repress EMT by Mdm2-mediated Slug degradation, and mutant p53 leads increased cancer cell invasiveness (Nat Cell Biol. 2009 Jun;11(6):694-704, PMID: 19448627). Our data showed that increased p53 signaling in R group, which was consistency with low EMT gene signature in R group. These comments are now included in the Result and Discussion section of the revised manuscript (p. 12, 17) and Extended Data Figure 10a, b.

2. By the way, a list of the used cell lines has not been provided; details of the experiments are missing as well as the partitioning of the cell lines between responders and non responders. This would not allow other researchers to confirm the data.

: For cell line drug response prediction, we used CCLE RPKM data for prediction. When we reviewed the expressions of predictor genes in the cell-lines with RNA-seq FPKM data obtained from Cell Model Passports of Sanger Institute, which is a higher quality data, we found that a large number of genes for non-responder prediction were not expressed. Since cell lines do not have tumor-microenvironment, it was reasonable that non-responder predictor genes, including many cell-to-cell interaction genes, were not expressed in cell lines. So, we have removed the cell-line prediction data from throughout the manuscript.

3. The way response has been evaluated in PDXs is questionable. To make a clinical comparison, response should be evaluated in relation to the volume at treatment start. This would allow to define progressive disease, stable disease and partial/complete response as it happens in patients.

: To evaluate the drug responsiveness in PDX models, we have used a classical method using treatment over control ratio. As reviewer recommended, we have tried to apply modifying RECIST

(Response Evaluation Criteria In Solid Tumors) for PDX tumor growth evaluation, which is largely applied in response assessing of neoadjuvant tumors. There were two methods used for mouse clinical trial assessment based on RECIST. William *et al.* assesses response by tumor volume change compared to the baseline tumor volume and makes individual mouse judgements (In: Patient-Derived Xenograft Models of Human Cancer. Springer International Publishing; 2017:141–154). Gao *et al.* makes treatment group-based judgements, picking a best response throughout the group and the observation time (day \geq 10) (Nat Med 2015;21:1318-25, PMID: 26479923). However, when applied to our data, both methods determined all PDX tumors into progressive disease, which is not consistent with clinical data.

We agree that the tumor response should be evaluated in relation to the volume at treatment start for clinical application. However, several co-clinical trials with PDX models also adopted the percentage tumor growth inhibition to evaluate the responsiveness (Ann Oncol. 2017, 28(6):1250-1259, PMID: 28460066; Clin Cancer Res. 2012, 18(24):6658-67, PMID: 23082000). To evaluate the response more strictly, we combined two criteria for the responsiveness: (1) drug treatment significantly inhibited tumor growth [two-way analysis of variance (ANOVA) $P < 0.001$ for R group, $P > 0.05$ for NR group]; and (2) the final tumor volume in drug-treated mice was less than half that in vehicle-treated mice (percent tumor growth inhibition (%TGI) $\geq 50\%$ for R group, %TGI $< 50\%$ for NR group) (Supplementary Table S3). The intermediate group consist of samples that did not meet these criteria. Our criteria in PDX models were significantly concordant with recurrence after adjuvant chemotherapy in matched patients ($P = 0.033$, Chi-square test), suggesting that our criteria could be applied for the response evaluation in PDX models. We modified the response criteria more clearly in the Methods section of the manuscript (p. 21).

Reviewer #2:

1. Can the authors provide any data regarding genomic stability of the PDXs, in terms of driver gene profiles or clonality profiles over repeated passage?

: We have not tried to generate the sequenced data over repeated passage. To confirm the genomic stability of the PDX models, we compared the paired data of patient and PDX tumors more precisely. In Extended Data Figure 2, we showed clonal architectures maintained over xenotransplantation. Although cellular prevalence of tumor cells increased in both R and NR PDX tumors with the microenvironment replacement (Extended Data Figure 2d), the median percentage of shared somatic mutation of 67% between patient and PDX tumors and median Pearson correlation of somatic mutation allele frequency of 0.72 ($P = 0.00016$). With oncogene annotation of OncoKB

(www.oncokb.org), mutations were annotated as likely oncogenic/oncogenic mutations, and 90% (338 of 376) of likely oncogenic/oncogenic mutations were shared between patient and PDX tumors. When we further compared somatic mutations with therapeutic importance (OncoKB levels of 1 to 4), mutations in genes applicable to FDA-approved drugs and standard care (level 1 and 2) were maintained in all of patient and PDX pairs. Full list of likely oncogenic/oncogenic mutations annotated by OncoKB was added to Supplementary Table 2 and a figure of druggable mutations below was added to Extended Data Figure 1. These comments are now included in the Result section of the revised manuscript (p. 5).

Extended Data Fig. 1.

T: patient tumor. X: PDX tumor. Black: Both patient and PDX tumors shared somatic mutations. Blue: Found in PDX tumor. Red: Found in patient tumor. Level 1: FDA-approved drugs, Level 2: Standard care, Level 3: Clinical evidence, Level 4: Biological evidence.

2. In Fig. 1b, why are three NRs chosen amongst the ‘intermediate’ range of PDXs? Likewise, why is one ‘intermediate’ PDX intermixed among the Rs?

: To evaluate the response more strictly, we combined two criteria for the responsiveness : (1) drug treatment significantly inhibited tumor growth [two-way analysis of variance (ANOVA) $P < 0.001$ for R group, $P > 0.05$ for NR group]; and (2) the final tumor volume in drug-treated mice was less than half that in vehicle-treated mice (percent tumor growth inhibition (%TGI) $\geq 50\%$ for R group, %TGI $< 50\%$ for NR group) (Supplementary Table S3). The intermediate group consist of

samples that did not meet these criteria. Three NR samples showed P values larger than 0.05 and one R sample showed P value lower than 0.001. We modified the response criteria more clearly in the Methods section of the manuscript (p. 21).

3. When analyzed across the entire PDX cohort, is there an association between tumor mutation burden, MMR mutation signatures, and the observed downregulation of MMR essential genes?

: Across 13 responder and 11 non-responder PDX tumors, tumor mutation burden, defective MMR signatures and *MLH1* expression levels were compared. Correlation coefficients are described in the table below. Mutation burden was highly correlated with defective MMR signature, and *MLH1* expression showed significant anti-correlation with mutation burden, and defective MMR signature. These data suggest that tumor mutation burden, MMR mutation signatures, and the expression of MMR essential gene were highly associated. These comments are now included in the Result section of the revised manuscript (p. 8) and Extended Data Figure 6c.

	dMMR signature (ID)	Mutation rate	MLH1 log2FPKM
dMMR signature (SBS)	0.924**	0.837**	-0.799**
dMMR signature (ID)		0.867**	-0.893**
Mutation rate			-0.751*

* $P < .0001$

** $P < .00001$

4. In Fig. 2A, 9 of the PDXs are designated as MSI-positive, which seems (to this Reviewer) rather high. The increase in TMB among these tumors relative to the non-MSI PDXs also does not seem that high. Can the authors confirm the MSI status of these, perhaps using the standard microsatellite panel of MLH1/MSH2 IHC?

: In our cohort, total 11 out of 31 patients (35.5%) were evaluated as MSI-high. We already analyzed samples using PCR and fragment analysis for five markers (BAT-25, BAT-26, D2S123,

D5S346, and D17S250), and the samples were regarded as MSI-high when the instability were detected in two or more markers. Depending on the analysis method, the reported frequency of MSI-high cases is highly variable in gastric cancer: 11.7% to 33.8% in Asian and 16.3% to 25.2% in European (Medicine 2017, 96(25):e7224, PMID: 28640116). In addition, high frequency of MSI-high was reported in intestinal-type gastric cancer in Korean patients (Korean J Intern Med. 2005, 20(2): 116–122, PMID: 16134765), suggesting that there is ethnic difference in the frequency of MSI-high gastric cancer patients. The proportions of MSI samples were 23% in the ACRG cohort, 16% in the Singapore-Duke study and 22% in the TCGA study. Previous exome analysis of 55 Korean gastric cancer patients (Proc Natl Acad Sci U S A 2015, 112:12492-7; PMID: 26401016) reported that the average tumor mutation burden was 19.58 mutations/Mb in MSI-H patients, and 5.91 mutations/Mb in MSS patients. TCGA-stomach cancer study showed average mutation rates of 50 in MSI samples and 5 in non-MSI samples (Nature 2014, 513:202-9; PMID: 25079317). In our PDX cohort, tumor mutation burden was average 39.8 mutation per Mb in MSI-H tumors and 6.0 in MSS tumors, which are consistent with previous reports.

5. In order to pinpoint differentially regulated genes between R and NR, the authors integrate transcriptomic data from both PDX and matched human tissues. This raises the question of whether the findings highlighted (for example, the enrichment of ECM-matrix in NR) might have been determined by analysis of human tumors alone, without relying on PDXs. Can the authors demonstrate or clarify the value of using the PDX system (also given that there is a correlation between patient response and PDX response).

: PDXs are increasingly used as drug development tools in pre-clinical settings and have been shown to recapitulate the histology and behavior of the cancers from which they are derived. However, whether PDXs mimic the clinical outcomes of patients remains controversial. Our data demonstrated that the estimated responsiveness to 5-FU and oxaliplatin-based chemotherapy in PDX models was highly correlated with clinical outcomes in the adjuvant setting (Fig. 1d-f) and that the prediction model, developed from analyses of PDX tumors, was able to predict patients' drug responsiveness and prognosis (Fig. 5). Therefore, our data suggest that PDX models could be applied for the development of prediction model for treatment responsiveness. There are some limitations in analyzing human samples compared to PDX models. Sometimes it is difficult to obtain large quantity of cancer tissues from patients, and tumor cell purity in tissues are more diverse in human samples than in PDX tumors. Patients' general conditions and environmental factors are hard to normalize in human study, which requires large size of patient cohort. Therefore, adopting PDX

models for the development of prediction model for drug response can reduce the development cost. In addition, separate analysis of tumor cells and stromal cells can be applied in PDX models by separately analyzing the sequencing reads from human and mouse, similarly with our approach. These comments are now included in the Discussion section of the revised manuscript (p. 19).

6. It is not clear to me why, in formulating their 29-gene model, the authors choose to include MLH1. This seems quite arbitrary to me and is not consistent with the unbiased data-driven approach used to select the other genes.

: We agree with reviewer's concern of selection criteria of classifier genes. Since MSI status and other defective mismatch repair signatures did not showed big difference, we re-developed our prediction model with unbiased data-driven approach based on significantly differently expressed genes (DEGs).

First, we have constructed 123 DEG-based models with multiple prediction algorithm: Those include DLDA, 1-Nearest Neighbor, 3-Nearest Neighbor, Nearest Centroid, Support Vector Machine, Random Forest, and Bayesian Compound Covariate Predictor (CCP). Bayesian CCP model showed the best performance (sensitivity for R: 0.86, specificity: 0.89, positive predictive value: 0.92, negative predictive value: 0.80). New 123-gene Bayesian CCP model showed a significant prognostic value in TCGA-stomach cancer cohort, yielding a log-rank p-value of 0.0062 in disease-free survival.

We further narrowed down to a 30-gene model by selecting 15 high-weighted genes for each responder and non-responder to build a model which maintains a predictive value of 123-DEG model and achieves clinical utility. Performance was similar with those of 123-gene model (sensitivity for R: 0.86, specificity: 0.89, positive predictive value: 0.92, negative predictive value: 0.80).

Previous molecular subtyping of gastric cancer required hundreds of genes, and it is less likely to be practical in clinic. In addition, previous subtyping cannot be sufficient for prognostic or therapeutic reference. Our 30-gene model can be more practical to make therapeutic decision. We have revised Results section in the manuscript (p.14), Figure 5 and Extended Data Figure 12,13,14,15, and 16.

7. As mentioned in the Discussion, several studies in gastric cancer have pointed to the existence of a ‘mesenchymal’ subtype of gastric cancer (Lei et al., 2013; ACRG study (MSS/EMT); Oh et al., 2018). To what extent does the author’s 29 gene signature overlap with these in a multivariate analysis to provide independent predictive power?

: As the reviewer indicated, several studies pointed the mesenchymal subtype of gastric cancer. ACRG used 310 genes to separate EMT subtype, and Oh *et al.* used 295 genes for prognostic expression signature. When we compared 123 differentially expressed genes (DEGs) between R and NR in our study with ACRG study and Oh *et al.*, total 43 of 123 genes were overlapped as below. Venn diagram was added in Extended Data Figure 10a. Several overlapping genes were associated EMT/mesenchymal types, and seven genes (*COL1A2*, *COL3A1*, *DKK3*, *ANTXR1*, *CMTM3*, *DPYSL3* and *HTRA1*) were common in both studies (Genes in blue indicate the genes associated with EMT).

Our 123 DEGs are probably a mixture of markers that differentiate molecular subtypes (for subtypes with poor prognosis, such as mesenchymal type) and markers related to FOLFOX responsiveness. However, compared to previous molecular subtype markers, the proportion of overlapping marker genes is about one third (43 of 123), and the remaining two thirds of DEGs might contain independent predictive markers related to drug responsiveness that have not been discovered in previous studies. These issues are now included in the Result section of the revised manuscript (p.12) and Extended Data Figure 10a.

8. Can the authors definitely address whether or not the 29-gene signature is truly predictive or also prognostic in nature? This could be addressed by testing it in a cohort of untreated gastric tumors.

: In TCGA stomach cohort, 61 of 261 patients were not treated with adjuvant chemotherapy (chemotherapy history of most patients are unavailable). Of 300 ACRG gastric cancer patients, 80 patients received adjuvant chemotherapy and 219 patients did not.

Survival analysis of untreated patients was performed with a 30-gene response prediction model. Disease free survival (DFS) was not significantly different between responder-like and non-responder-like groups in a TCGA cohort as below. However, in the ACRG cohort, patients who were

reassigned to responder-like showed significant better prognosis (log-rank P-value < 0.0001). There should be an extension of research to find out if our classifier genes are prognostic or predictive.

Reviewers' Comments:

Reviewer #1:

Remarks to the Author:

1) I still do not see the novelty related to MMR. The authors did not find any difference in MLH1 expression, MMR or tumor mutation burden between R and NR. They describe that mutation burden was highly correlated with defective MMR signature, while MLH1 expression was anticorrelated with mutation burden and defective MMR. This is not novel at all.

2) It is quite disturbing that all the cell line prediction data have been removed from the manuscript.

3) I am convinced that the way the PDX experiments have been shown does not allow the reader to fully interpret the results and define the quality of the response. How is it possible to define a threshold to define what is R and what is NR? A growth curve of the tumors would have been much simpler and clear, showing the effect of the treatment in terms that recapitulate what is observed in patients.

Reviewer #2:

Remarks to the Author:

The authors have responded well to my original concerns.

Responses to Reviewer's comments:

Below please find our responses to each specific comment and suggestion raised by the reviewers.

Reviewer #1 (Remarks to the Author):

1) I still do not see the novelty related to MMR. The authors did not find any difference in MLH1 expression, MMR or tumor mutation burden between R and NR. They describe that mutation burden was highly correlated with defective MMR signature, while MLH1 expression was anticorrelated with mutation burden and defective MMR. This is not novel at all.

: We agree that we did not find any novel finding related with MMR in the drug responsiveness. We only suggested the possible role of a defective MMR pathway in the responsiveness of 5-FU-based adjuvant chemotherapy. Our data showed a higher frequency of MSI-H (R: 38.5%, NR: 27.3%) and following enrichment of deficient MMR mutational signatures in responders (Fig. 2b, c), but it is difficult to say that those features are general characteristics of the responder group. So, we toned down the significance of defective DNA MMR pathway in drug responsiveness and removed this issue from main points of our findings in the Discussion section. But we described this issue in the separate paragraph of Discussion section of the revised manuscript (p. 17).

2) It is quite disturbing that all the cell line prediction data have been removed from the manuscript.

: We are sorry for the change in cell line data. During the revision process, we reviewed the expressions of predictor genes in the cell-lines with RNA-seq FPKM data obtained from Cell Model Passports of Sanger Institute, which is a higher quality data, we found that a large number of genes for non-responder prediction were hardly expressed. Since cell lines do not have tumor-microenvironment, it is reasonable that non-responder predictor genes, including many cell-to-cell interaction genes, are not expressed in cell lines. So, we have removed the cell-line prediction data from throughout the manuscript. However, we validated our prediction model using human gastric cancer cohort of ACRG, TCGA and Singapore-Duke study (Fig. 5).

3) I am convinced that the way the PDX experiments have been shown does not allow the reader to fully interpret the results and define the quality of the response. How is it possible to define a threshold to define what is R and what is NR? A growth curve of the tumors would have been much simpler and clear, showing the effect of the treatment in terms that recapitulate what is observed in patients.

: As the reviewer indicated, it is ambiguous to define a threshold for responder and non-responder in mouse tumor models. Nonetheless, to generate the prediction model for drug responsiveness, we need to apply specific criteria to determine the drug responsiveness. Comparing relative tumor volume of final and starting point in each single tumor is one of several ways to determine drug efficacy in the PDX models. As shown in the Method: *In vivo* pharmacological studies section, we have jointly applied two criteria to determine the responsiveness of PDX tumors. Responders met both criteria below.

We (1) assessed potential differences in tumor volume between treated and vehicle arms with statistical method using ANOVA ($P < 0.0001$ for responders, $P > 0.05$ for non-responders) and (2) quantified drug efficacy with Treated/Control (T/C) ratio ($< 50\%$ for responders; Data was supplied in Supplementary Table S3). Growth curves of PDX tumors was already shown in the Supplementary Figure 1.

Each ANOVA and T/C ratio method is widely used in pre-clinical setting. While reviewing, we found tumor growth inhibition (TGI (%)) rate is better suited for comparing relative tumor volume changes and providing a more objective threshold criterion than T/C ratio.

$$TGI (\%) = \frac{TV_{vehicle} - TV_{treatment}}{TV_{vehicle} - TV_{initial}} \times 100$$

$TV_{vehicle}$: the tumor volume for the vehicle-treated animals at a specified endpoint time

$TV_{initial}$: the initial tumor volume at the start of the treatment

$TV_{treatment}$: the tumor volume of the treatment groups at a specified endpoint time

1. Wong et al. (Clin Cancer Res. 2012 Jul 15;18(14):3846-55, PMID: 22648270) found out that agents, which led to greater than 60% TGI in preclinical models at clinically relevant exposures, are more likely to lead to responses in the clinic.
2. Gurard-Levin et al. (Mol Cancer Ther. 2016 Jul;15(7):1768-77, PMID: 27196757) and Chen et al. (J Hematol Oncol. 2018 Feb 13;11(1):20, PMID: 29433585) applied TGI method and determined high-responder (TGI $> 60\%$) and poor-responder (TGI $< 30\%$) in drug efficacy evaluation using breast and gastric cancer PDX models,

respectively.

All of 13 responder PDX tumors in our study showed TGI greater than 70% as below (ANOVA P -value was displayed together. **: ANOVA $P < 0.0001$, *: $0.0001 \leq P < 0.05$).

Since there were significantly consistent responses to 5-FU and oxaliplatin-based chemotherapy between matched PDX models and patients (Figure 1c-e), our results are in line with the observations in Wong et al. (Clin Cancer Res. 2012 Jul 15;18(14):3846-55, PMID: 22648270).

We revised the data by applying the responsiveness criteria using TGI method and modified the Fig. 1b, and Supplementary Table 3. The TGI methods were described in the Results (p. 6) and Methods section of the revised manuscript (p. 21).

Reviewers' Comments:

Reviewer #1:

Remarks to the Author:

I regret that both the observations I did, led to the removal of part of the work.